# TRAC: Tensor-Train based Across-layer Compression for Parameter-Efficient Fine-Tuning

**Bangguo Ye**[1], **Yuanwei Zhang** [1], **Xiaoqun Zhang**[1, 2, 3] *

[1] School of Mathematical Sciences, Shanghai Jiao Tong University, Shanghai, China
[2] Shanghai-Chongqing Institute of Artificial Intelligence, Chongqing, China
[3] Institute of Natural Sciences, Shanghai Jiao Tong University, Shanghai, China
{bgye, sjtuzyw, xqzhang}@sjtu.edu.cn

## Abstract

Fine-tuning large pre-trained models under resource constraints remains challenging due to the massive number of parameters involved. Existing parameter-efficient tuning methods, such as low-rank adaptation (LoRA) and its variants, rely heavily on matrix factorization and often struggle in extremely low-parameter regimes. In this work, we propose TRAC, a novel fine-tuning framework that leverages **T**ensor-**Tr**ain decomposition with **A**cross-layer **C**ompression. Specifically, TRAC represents each adaptation module as a compact sequence of tensor-train cores and allows certain cores to be frozen or shared across layers, thereby exploiting the inherent similarity and redundancy among layer weight matrices. To retain layer-specific flexibility, lightweight controllers are introduced, enabling shared tensor cores to adaptively modulate representations. We evaluate TRAC on diverse architectures, including Qwen, LLaMA, GPT, BERT, and ViT, across benchmarks covering text classification, text generation, and image classification. Experimental results demonstrate that TRAC achieves performance comparable to or better than LoRA and its variants, while substantially reducing trainable parameters and storage requirements. [1]

## 1 Introduction

Large pre-trained models have demonstrated remarkable performance across natural language processing, computer vision, and multimodal tasks (Zhao et al., 2023; Khan et al., 2022; Yin et al., 2023). However, their ever-increasing size presents significant challenges for fine-tuning and deployment on resource-constrained devices. To address this, parameter-efficient fine-tuning (PEFT) approaches (Houlsby et al., 2019; Li & Liang, 2021; Hu et al., 2022; Ding et al., 2023b) have been developed. Notably, low-rank adaptation (LoRA) (Hu et al., 2022) has become prominent due to its simplicity and efficiency in updating only a small fraction of parameters (Mao et al., 2025).

Despite its effectiveness, LoRA leaves substantial room for further compression. Its parameter count is bounded by matrix factorization, which establishes a minimum footprint tied to the discrete rank $r$. As model sizes grow and service providers may need to store millions of user-specific LoRA weights, even this reduced parameter count becomes non-trivial (Kopiczko et al., 2024). Moreover, the dependence on rank limits the granularity of parameter configurations, restricting LoRA's applicability in extremely low-parameter regimes (Koohpayegani et al., 2024). These limitations highlight the need for approaches beyond matrix factorization.

Recent research has explored reducing LoRA's parameter count by introducing parameter freezing or sharing across layers, at either the vector or matrix level (Karimi Mahabadi et al., 2021; Kopiczko et al., 2024; Gao et al., 2024; Li et al., 2024; Liu et al., 2024b; Song et al., 2024). These approaches exploit the insight that LoRA matrices across different layers exhibit similarities and redundancies, allowing further compression beyond standard low-rank adaptation. However, they remain confined to the matrix decomposition paradigm, whose inherent structural constraints make it difficult to reach extremely small parameter budgets.

---

*Corresponding Author
[1]Code is available at `https://github.com/BangguoYe/TRAC`.

In parallel, tensor decomposition frameworks have recently been introduced for parameter-efficient fine-tuning (Liu et al., 2021; Jie & Deng, 2023; Bershatsky et al., 2024; Yang et al., 2024; Hounie et al., 2024; Chen et al., 2024). Compared with matrix factorization, tensor decompositions are more expressive and flexible, supporting a broader range of parameter configurations while maintaining strong representational power. Yet, most existing tensor-based approaches, such as LoRETTA (Yang et al., 2024) and QuanTA (Chen et al., 2024), treat every decomposition module as an independent set of trainable tensors. By not fully exploiting cross-layer redundancies, these methods incur parameter growth that scales with model depth, thereby limiting their efficiency gains.

Motivated by these complementary gaps, we propose TRAC (**T**ensor-**Tr**ain based **A**cross-layer **C**ompression), a new fine-tuning framework that unifies the strengths of the above two directions. TRAC represents each adaptation module as a compact sequence of Tensor-Train cores (Oseledets, 2011), while strategically freezing and sharing certain cores across layers to capitalize on redundancy among layer weight matrices. To maintain layer-specific flexibility, lightweight controllers are introduced to adaptively modulate the shared tensor cores. This design enables TRAC to achieve extremely low trainable parameter counts while preserving strong performance.

Our contributions can be summarized as follows:

1. We propose TRAC, a novel PEFT framework that integrates Tensor-Train decomposition with cross-layer compression. TRAC substantially reduces trainable parameters while preserving or improving performance compared to prior methods.

2. We leverage cross-layer weight redundancy by sharing Tensor-Train cores, enabling parameter savings beyond standard matrix-decomposition-based PEFT. We validate the effectiveness of our design through controlled ablations and empirical comparisons.

3. Across diverse benchmarks, TRAC demonstrates competitive or superior performance compared to LoRA and related variants while using significantly fewer parameters. TRAC achieves up to a $20\times$ reduction in trainable parameters for LLaMA2-13B on SuperGLUE and up to a $14\times$ reduction for ViT-Large on image classification tasks, while maintaining or improving accuracy.

## 2 RELATED WORK

**LoRA and Its Variants.** Low-Rank Adaptation (LoRA) (Hu et al., 2022) maintains a frozen pretrained weight matrix $W_0 \in \mathbb{R}^{m \times n}$ while expressing its update $\Delta W$ through a low-rank factorization:

$$h = W_0 x + \Delta W x = W_0 x + BA^\top x, \tag{1}$$

where $B \in \mathbb{R}^{m \times r}$ and $A \in \mathbb{R}^{n \times r}$ are low-rank matrices with rank $r \ll \min(m, n)$. This elegant design reduces the number of trainable parameters while preserving the frozen backbone. However, the decomposition structure imposes a lower bound on trainable parameters, reached at $r = 1$, which limits its applicability in *extremely* parameter-constrained settings. Subsequently, a rich line of research has emerged on pushing LoRA variants into more efficient regimes (Mao et al., 2025).

**Tensor-Based Fine-Tuning Approaches.** Beyond matrix factorization, tensor decompositions provide a broader framework for parameter compression. Classical methods include Canonical Polyadic (CP) (Hitchcock, 1927), Tucker (Tucker, 1966), and Tensor-Train (TT) (Oseledets, 2011), each with different representational trade-offs (Grasedyck et al., 2013). Recent works have explored these directions for fine-tuning large models. For example, LoRTA (Hounie et al., 2024) applies a CP-style low-rank tensor parametrization to further reduce LoRA parameters, while LoTR (Bershatsky et al., 2024) adopts a Tucker-inspired framework. However, CP faces the classic difficulties of rank selection and numerical instability, whereas Tucker requires a dense core tensor whose size grows exponentially with order, which limits its scalability to high-order tensors.

Among the concurrent developments, many studies have adopted TT decomposition. By factorizing matrices into sequences of compact third-order cores, TT achieves linear parameter growth and allows explicit control of compression through TT-ranks. Representative TT-based approaches illustrate these advantages: LoRETTA (Yang et al., 2024) applies TT decomposition directly to the low-rank matrices within LoRA; TT-LoRA (Anjum et al., 2024) instead represents the incremental update matrices $\Delta W$ in TT format; and QuanTA (Chen et al., 2024) leverages quantum-inspired TT structures for efficient high-rank adaptation. In line with these developments, our method is built upon the TT framework as the basis for parameter-efficient fine-tuning.

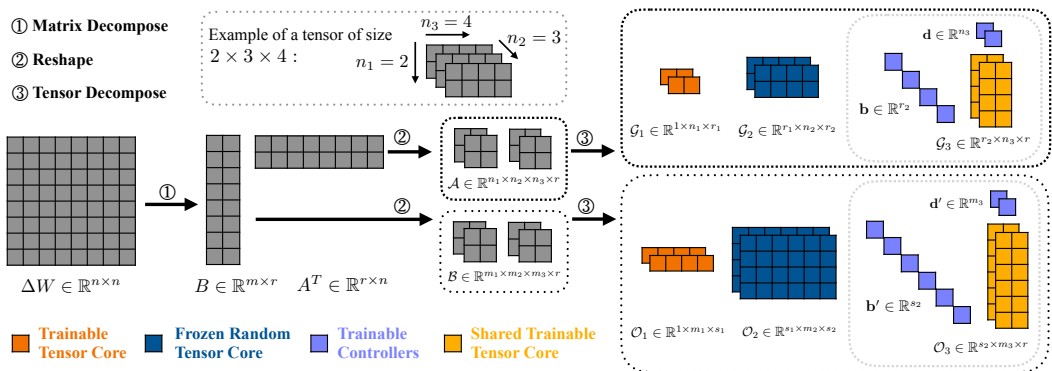

Figure 1: Overview of our proposed method. TRAC reconstructs LoRA adaptation modules into Tensor-Train form, represented by a mixture of trainable, frozen, and shared tensor cores. Lightweight controllers are introduced to enhance the expressiveness of shared cores across layers. TRAC provides flexible control over parameter budgets by adjusting TT ranks, allowing, for example, more resources to be allocated to matrix $B$ (Zhu et al., 2024).

**Random or Shared Weights in Fine-Tuning.** A complementary research direction exploits frozen or shared structures rather than explicit tensorization. Early analyses highlighted the expressiveness of randomly initialized, frozen subnetworks (Frankle & Carbin, 2019; Ramanujan et al., 2020), while weight sharing exhibited effectiveness in compact Transformer designs (Lan et al., 2020). Recent PEFT methods extend these insights into LoRA-style tuning: VeRA (Kopiczko et al., 2024) shares LoRA matrices across layers while training only scalars; NoLA (Koohpayegani et al., 2024) relies on random matrices combined via learned coefficients; FourierFT (Gao et al., 2024) transforms updates into the Fourier domain for cross-layer sharing of spectral components; VB-LoRA (Li et al., 2024) trains a global pool of shared vectors to assemble low-rank adapters; AFLoRA (Liu et al., 2024b) progressively freezes LoRA matrices $A$ and $B$ during later training; ShareLoRA (Song et al., 2024) enforces systematic cross-layer sharing of LoRA matrices; and COMPACTER (Karimi Mahabadi et al., 2021) inserts task-specific adapter weights summing Kronecker products of shared weights and per-layer rank-one matrices. Collectively, these strategies demonstrate that randomization and sharing, when carefully controlled, can offer strong parameter efficiency and highlight the potential of exploiting cross-layer redundancy—a principle our method builds upon.

## 3 METHODOLOGY

When scaling to very large models, existing LoRA variants and tensor-based PEFT methods still impose notable parameter costs, limiting their applicability in severely resource-constrained scenarios. To address this, we propose **TRAC** (**T**ensor-**Tr**ain based **A**cross-layer **C**ompression), a parameter-efficient fine-tuning framework that combines Tensor-Train decomposition with across-layer sharing and freezing of selected tensor cores, enhanced by lightweight controllers (Figure 1). In practice, each LoRA module is replaced by a compact tensor-train structure consisting of a trainable core, a frozen core, and a shared core modulated by the controllers, providing both efficiency and flexibility. This design exploits cross-layer redundancy to substantially reduce trainable parameters while preserving adaptability, enabling TRAC to achieve highly compact yet expressive fine-tuning modules.

### 3.1 TENSOR-TRAIN DECOMPOSITION AND TENSORIZED LAYER

Tensor-Train (TT) decomposition (Oseledets, 2011) is widely employed for managing high-dimensional data. For a given $d$-order tensor $\mathcal{X} \in \mathbb{R}^{n_1 \times \cdots \times n_d}$, the Tensor-Train format expresses $\mathcal{X}$ as a collection of $d$ three-order tensors $\mathcal{G}_k \in \mathbb{R}^{r_{k-1} \times n_k \times r_k}$, $k = 1, \ldots, d$, with rank $(1, r_1, \ldots, r_{d-1}, 1)$. Each tensor $\mathcal{G}_k$ is referred to as the $k$-th *tensor core*. The elements of $\mathcal{X}$ can be represented in the following matrix-product form:

$$\mathcal{X}(i_1, \ldots, i_d) = \mathcal{G}_1(i_1) \cdots \mathcal{G}_d(i_d), \tag{2}$$

where $\mathcal{G}_k(i_k) \in \mathbb{R}^{r_{k-1} \times r_k}$ is the $i_k$-th slice of $\mathcal{G}_k$ along the second mode. Assuming $r_k = r$, $n_k = n$, then the effective dimension of $\mathcal{X}$ in (2) is $dnr^2$, which scales linearly with tensor order, thus

Table 1: Comparison on RoBERTa-Base between the All Trainable baseline and our freeze-share strategy under identical TT-rank. Mean and standard deviation are reported.

| Method | # Train. Param. | CoLA | MRPC | RTE | STS-B |
|---|---|---|---|---|---|
| All Trainable | 411,648 | 64.83(2.21) | 88.92(0.80) | 80.29(1.91) | 91.21(0.19) |
| Ours (Freeze+Share) | **26,304** | 64.84(2.89) | 88.63(1.79) | 80.14(1.80) | 91.21(0.20) |

achieving an exponential compression compared to the full ambient dimension $n^d$. These properties render Tensor-Train decomposition highly efficient for representing high-order and high-dimensional data, widely applied in quantum physics (Lanyon et al., 2017), recommendation systems (Zhou et al., 2024), and many others (Kolda & Bader, 2009; Zhang et al., 2025).

We also define notation for the tensorization of matrices. For a matrix $W \in \mathbb{R}^{m \times n}$ with factorizations $\boldsymbol{m} = (m_1, \ldots, m_{d_m})$, $\prod_{i=1}^{d_m} m_i = m$ and $\boldsymbol{n} = (n_1, \ldots, n_{d_n})$, $\prod_{i=1}^{d_n} n_i = n$, we denote the reshape operator as

$$\mathcal{W} = \text{Reshape}(W, \boldsymbol{m}, \boldsymbol{n}),$$

where the reshaped tensor $\mathcal{W}$ is of order $(d_m + d_n)$ with size $(\boldsymbol{m}, \boldsymbol{n})$.

## 3.2 Baseline: Training All Tensor Cores

Motivated by the need to compactly represent the large parameter increment matrix $\Delta W$ in (1), we employ TT format for parameter compression. Given LoRA's low-rank factorization $\Delta W = BA^\top$ with $A \in \mathbb{R}^{n \times r}$, $B \in \mathbb{R}^{m \times r}$, one straightforward approach is to apply TT decomposition directly on $A$ and $B$ and train all tensor cores. We refer to this approach as the *All Trainable* baseline.

For instance, for $A$ we consider:

$$\mathcal{A} = \text{Reshape}(A, (n_1, n_2, n_3), r), \tag{3}$$

with $\mathcal{A} \in \mathbb{R}^{n_1 \times n_2 \times n_3 \times r}$, $n_1 n_2 n_3 = n$. Following the TT representation format

$$\mathcal{A}(i_1, i_2, i_3, :) = \mathcal{G}_1(i_1)\mathcal{G}_2(i_2)\mathcal{G}_3(i_3) \tag{4}$$

with rank $(1, r_1, r_2)$ and cores $\mathcal{G}_1 \in \mathbb{R}^{1 \times n_1 \times r_1}$, $\mathcal{G}_2 \in \mathbb{R}^{r_1 \times n_2 \times r_2}$, $\mathcal{G}_3 \in \mathbb{R}^{r_2 \times n_3 \times r}$, and all $\mathcal{G}_i$ are trainable. An analogous decomposition is applied for $B$.

This strategy is similar in spirit to earlier tensorized PEFT methods such as LoRETTA (Yang et al., 2024) and TT-LoRA (Anjum et al., 2024). Compared with vanilla LoRA, it reduces parameters due to TT compression, yet the total parameter count still scales linearly with network depth since each layer has its own independent set of tensor cores.

## 3.3 Limitations Observed: Redundancy Across Layers

Among the three cores in the TT representation, $\mathcal{G}_1$ is relatively small because it includes a dimension of size 1, yielding parameter count $|\mathcal{G}_1| = n_1 r_1$. In contrast, $\mathcal{G}_2$ and $\mathcal{G}_3$ are much larger with sizes $|\mathcal{G}_2| = r_1 n_2 r_2$ and $|\mathcal{G}_3| = r_2 n_3 r$, and thus dominate the parameter budget. To reduce redundancy, our strategy keeps $\mathcal{G}_1$ trainable in each layer while freezing $\mathcal{G}_2$ and sharing $\mathcal{G}_3$ across layers.

Although the All Trainable baseline reduces parameters compared to LoRA, we empirically found that in deep transformers many layers exhibit redundant parameterization, especially within the large cores $\mathcal{G}_2, \mathcal{G}_3$. To corroborate this, we conducted controlled experiments on RoBERTa-Base (Liu, 2019), which consists of a 12-layer Transformer encoder with hidden size 768. Following common practice, we applied adaptation only to the self-attention $W_Q$ and $W_V$ matrices, each of shape $768 \times 768$. Both the All Trainable baseline and our freeze-share scheme employed the same tensor shape $[8, 8, 12]$ and TT-rank $[16, 24]$ with LoRA rank $r = 16$, leading to

$$\mathcal{G}_1 \sim [1, 8, 16], \ \mathcal{G}_2 \sim [16, 8, 24], \ \mathcal{G}_3 \sim [24, 12, 16].$$

As shown in Table 1, even though the All Trainable baseline introduces nearly $15\times$ more parameters than our freeze-share scheme (0.41M vs 0.026M), the performance improvement is marginal and often statistically comparable. This highlights the substantial redundancy across layers and supports our design choice of freezing $\mathcal{G}_2$ and sharing $\mathcal{G}_3$ while keeping $\mathcal{G}_1$ layer-specific.

### 3.4 TRAC: Tensor-Train based Across-layer Compression

Building upon the observations above, we propose **TRAC**. For each weight matrix $W$, we still represent LoRA matrices $A, B$ in TT format, but assign distinct roles to different cores:

$$\mathcal{G}_1 \text{ (trainable)}, \quad \mathcal{G}_2 \text{ (frozen)}, \quad \mathcal{G}_3 \text{ (shared across layers with lightweight controllers).} \quad (5)$$

Concretely, within the Transformer, one randomly initialized $\mathcal{G}_2, \mathcal{G}_3$ pair is allocated to each type of matrix (query, key, value, feed-forward). $\mathcal{G}_2$ is frozen, while $\mathcal{G}_3$ is shared across all layers of the same type. This sharing substantially reduces parameter usage while leveraging layer-wise redundancy.

To enrich the capacity of shared tensors, we introduce two lightweight vector *controllers* $\mathbf{b} \in \mathbb{R}^{r_2}$ and $\mathbf{d} \in \mathbb{R}^{n_3}$ per layer, gating the shared tensor $\mathcal{G}_3$ in a multiplicative manner:

$$\widetilde{\mathcal{G}_3}(t_1, t_2, t_3) = \gamma(\mathbf{b}, \mathbf{d}) \cdot \sigma(\mathbf{b})(t_1) \cdot \sigma(\mathbf{d})(t_2) \cdot \mathcal{G}_3(t_1, t_2, t_3), \quad (6)$$

where $\gamma$ is a scaling factor (fixed or trainable), and $\sigma(\cdot)$ is a task-dependent activation (we use exponential for text classification, linear for text generation and image classification). These controllers act as lightweight gating signals, akin to MoE-style expert weighting (Cai et al., 2025) or DARTS-style continuous operator selection (Liu et al., 2019), thereby enriching layer-specific expressivity.

After applying the controllers, the vector-wise expression of $\mathcal{A}$ is

$$\mathcal{A}(i_1, i_2, i_3, :) = \mathcal{G}_1(i_1)\mathcal{G}_2(i_2)\widetilde{\mathcal{G}_3}(i_3).$$

We then take a compact notation denoted as $A = [\mathcal{G}_1, \mathcal{G}_2, \mathcal{G}_3, \mathbf{b}, \mathbf{d}]$. For matrix $B \in \mathbb{R}^{m \times r}$, we similarly obtain $B = [\mathcal{O}_1, \mathcal{O}_2, \mathcal{O}_3, \mathbf{b}', \mathbf{d}']$, where $\mathcal{O}_1 \in \mathbb{R}^{1 \times m_1 \times s_1}, \mathcal{O}_2 \in \mathbb{R}^{s_1 \times m_2 \times s_2}, \mathcal{O}_3 \in \mathbb{R}^{s_2 \times m_3 \times r}, \mathbf{b}' \in \mathbb{R}^{s_2}, \mathbf{d}' \in \mathbb{R}^{m_3}$, and $(1, s_1, s_2)$ is the rank.

At inference, TRAC, similar to LoRA, merges with pretrained weights without altering architecture.

### 3.5 Implementation Details of TRAC

**Initialization.** The initialization of tensor cores is crucial for model performance. For a weight matrix $W \in \mathbb{R}^{m \times n}$ and its corresponding tensor core $\mathcal{G}_k \in \mathbb{R}^{r_{k-1} \times n_k \times r_k}$, we utilize an initialization formula adapted from the LoRETTA (Yang et al., 2024) implementation:

$$\sigma_k = \frac{1}{\sqrt{r_k}} \left( \frac{1}{m+n} \right)^{\frac{1}{2n_k}}. \quad (7)$$

We term this approach Tensor-Train normal initialization (TT Norm.), as it accounts for both tensor ranks and shape factors in the TT formulation. This initialization ensures variance stability in the reconstructed matrices, mitigating exploding or vanishing as tensor dimensions and ranks change.

Following LoRA convention (Hu et al., 2022), we initialize the last core in $B$ (i.e., $\mathcal{O}_3$) to zero, ensuring the initial $B$ matrix is a zero matrix. Vector controllers are initialized based on the activation function $\sigma(\cdot)$ to yield an identity operation at the start of training. We also evaluate alternative approaches like Kaiming initialization (He et al., 2015) in our ablation studies (Appendix A.1).

**Components.** TRAC consists of two key components: shared tensor cores for parameter efficiency and vector controllers for layer-specific adaptability. This design balances efficiency with performance, validated by ablation studies on BERT and ViT models as well as parameter-count analysis in Appendix A.2. These results confirm that both components are essential: training all tensor cores independently drastically increases parameters but yields only marginal gains, whereas adding controllers offers clear performance improvements with negligible extra cost.

**Asymmetry.** In our framework, it is convenient to accommodate the asymmetric property of $A$ and $B$ by assigning different tensor shapes and ranks for their representations. Here the asymmetry means that the importance of $A$ and $B$ may differ during fine-tuning. Previous research (Zhu et al., 2024) suggests that matrix $B$ is more influential and thus requires higher parameter capacity, which can be easily achieved in our approach by assigning larger tensor ranks. For example, we use ranks $(r_1, r_2)$ for $A$ and $(3r_1, 3r_2)$ for $B$ across tasks, which shows better performance than using $(2r_1, 2r_2)$ for both $A$ and $B$. Our detailed analysis in Appendix A.3 further confirms this finding, with experiments showing that allocating more parameters to matrix $B$ consistently yields better results.

Table 2: Orders of parameter and computational complexity for different fine-tuning methods.

|  | FT | LoRA | VeRA | LoRETTA | TRAC |
|---|---|---|---|---|---|
| PARAMETER ORDER | $O(Ln^2)$ | $O(Lnr)$ | $O(Ln)$ | $O(Ln^{1/3}r^2)$ | $O(n^{1/3}r^2)$ |
| COMPUTATIONAL ORDER | $O(Ln^2)$ | $O(Lnr)$ | $O(Lnr_{\text{VeRA}})$ | $O(Lnr)$ | $O(Lnr)$ |

**Hyperparameter Selection.** Compared with standard LoRA (whose main hyperparameters are rank $r$ and learning rate), our tensor-based method introduces two additional factors: tensor order and tensor ranks. A 3rd-order tensor is the minimal configuration that accommodates trainable, frozen, and shared cores; higher orders only add frozen intermediate cores, slightly increasing cost without improving accuracy. For ranks, the trainable parameter count depends on tensor ranks, model depth, and dimensions, allowing us to set $(r_1, r_2)$ under a target budget. See Appendix A.4 for details.

## 4 THEORETICAL ANALYSIS

### 4.1 APPROXIMATION AND SHARED STRUCTURES

We provide some theoretical analysis to understand why TRAC can achieve strong approximation ability under extremely compact parameter budgets. Our discussion is based on two key observations.

**Approximation Error via TT ranks.** As shown in classical results for Tensor-Train decompositions (Oseledets, 2011), the approximation error of TT-SVD is controlled by the neglected singular values in intermediate unfoldings. Informally, larger TT ranks yield smaller approximation errors, and exact recovery is possible once the TT ranks exceed the true unfolding ranks. While this property is well known in the general literature, we also derive in Appendix B.1 a specialized analysis for our setting, which makes the role of the specific tensorized LoRA modules explicit.

**Shared TT Structures Across Layers.** Consider a family of matrices $\{A^l\}_{l=1}^L$ with some sharing/redundancy, for example, where each can be factorized as $A^l = Q_0 R^l$, $l = 1, 2, \ldots, L$, with a common $Q_0$ and layer-specific, bounded $R^l$. In such case, the approximation error of jointly representing $\{A^l\}$ by sharing TT cores corresponding to $Q_0$ is bounded by a constant factor times the error of approximating $Q_0$. We provide a detailed formulation and derivation in Appendix B.2.

**Implication for TRAC.** Taken together, these two results indicate that (i) approximation quality increases with TT ranks, and (ii) partially sharing TT cores across layers can well control the error for families of matrices with shared structure. Consequently, *TRAC can afford to use larger tensor ranks under the same parameter budget*—thanks to across-layer sharing—thus achieving smaller approximation error in practice and explaining its empirical effectiveness.

### 4.2 PARAMETER EFFICIENCY ANALYSIS

We analyze the parameter efficiency of TRAC in comparison with other fine-tuning methods. Consider a transformer with $L$ layers and hidden dimension $m = n$. For LoRA, the parameter order is $O(Lnr)$; for VeRA it becomes $O(Ln)$ due to parameter sharing; for LoRETTA it is $O(Ln^{1/3}r^2)$; and for TRAC it is $O(n^{1/3}r^2)$ by sharing tensor cores across layers. Since $L$ and $n$ are typically large in modern models, TRAC has an asymptotic advantage compared to VeRA and LoRETTA. This parameter advantage, combined with the earlier analysis that larger TT ranks yield smaller errors, allows TRAC to use higher ranks under the same budget and achieve better approximation. Details are in Appendix B.3.

### 4.3 COMPUTATIONAL COMPLEXITY ANALYSIS

For a weight matrix $W \in \mathbb{R}^{n \times n}$ as defined in (1), we analyze the computational cost of $\Delta W x$, excluding operations common to all methods, where the *computational order* refers to the dominant term. Full fine-tuning requires $O(n^2)$ operations, and LoRA needs $O(nr)$ per layer. For VeRA, the effective rank $r_{\text{VeRA}}$ is typically very large (e.g., $r_{\text{VeRA}} \approx n$ in practice); thus its per-layer complexity approaches $O(nr_{\text{VeRA}}) \approx O(n^2)$. LoRETTA and TRAC can employ fast tensor-train multiplications with a dominant cost of $O(nr)$ per layer. Table 2 summarizes these complexities across the entire model with $L$ layers. Derivations are in Appendix B.4.

Table 3: Performance comparison of fine-tuning methods on the GLUE benchmark using RoBERTa Base and Large. We report Matthews correlation for CoLA, Pearson correlation for STS-B, and accuracy for other tasks. Higher values indicate better performance. Results marked with ([1]) are from (Hu et al., 2022).

| | Method | # Train. Param. | SST-2 | MRPC | CoLA | QNLI | RTE | STS-B | Avg. |
|---|---|---|---|---|---|---|---|---|---|
| BASE | FT[1] | 124.69M | 94.8 | 90.2 | 63.6 | 92.8 | 78.7 | 91.2 | 85.2 |
| | LoRA | 0.295M | **94.2** $_{\pm 0.1}$ | 88.0 $_{\pm 1.1}$ | 64.5 $_{\pm 0.6}$ | **92.6** $_{\pm 0.2}$ | **82.2** $_{\pm 0.9}$ | **91.8** $_{\pm 0.1}$ | **85.6** |
| | VeRA | 0.043M | 93.4 $_{\pm 0.3}$ | **89.2** $_{\pm 0.7}$ | 62.6 $_{\pm 0.7}$ | 90.5 $_{\pm 0.3}$ | 77.1 $_{\pm 1.7}$ | 91.0 $_{\pm 0.3}$ | 83.9 |
| | LoRETTA | 0.068M | 93.0 $_{\pm 0.5}$ | 87.7 $_{\pm 0.7}$ | 64.5 $_{\pm 2.8}$ | 90.5 $_{\pm 0.6}$ | 73.6 $_{\pm 2.7}$ | 89.7 $_{\pm 0.8}$ | 83.2 |
| | **TRAC** | **0.026M** | 93.7 $_{\pm 0.3}$ | 88.6 $_{\pm 1.8}$ | **64.8** $_{\pm 2.9}$ | 91.4 $_{\pm 0.4}$ | 80.1 $_{\pm 1.8}$ | 91.2 $_{\pm 0.2}$ | 85.0 |
| LARGE | FT[1] | 355.42M | 96.4 | 90.9 | 68.0 | 94.7 | 86.6 | 92.4 | 88.2 |
| | LoRA | 0.8M | **95.8** $_{\pm 0.3}$ | 88.1 $_{\pm 0.8}$ | **69.2** $_{\pm 1.3}$ | **94.7** $_{\pm 0.2}$ | **88.8** $_{\pm 1.5}$ | **92.5** $_{\pm 0.1}$ | **88.2** |
| | VeRA | 0.061M | 95.7 $_{\pm 0.4}$ | 88.0 $_{\pm 1.1}$ | 67.8 $_{\pm 1.6}$ | 93.9 $_{\pm 0.2}$ | 86.8 $_{\pm 1.9}$ | 92.3 $_{\pm 0.2}$ | 87.4 |
| | **TRAC** | **0.041M** | 95.4 $_{\pm 0.4}$ | **89.7** $_{\pm 1.1}$ | 68.2 $_{\pm 1.8}$ | 94.2 $_{\pm 0.1}$ | 87.3 $_{\pm 0.7}$ | 91.6 $_{\pm 0.4}$ | 87.7 |

# 5 EXPERIMENTS

We conducted extensive experiments on models ranging from 86M to 13B parameters across diverse benchmarks, covering Natural Language Understanding (NLU), Natural Language Generation (NLG), commonsense reasoning, mathematical reasoning, code generation, and image classification. Our results demonstrate the effectiveness of TRAC across different architectures and task types, providing strong empirical support for our method. Additional comparisons and extended experiments are included in Appendix C.

**Methods in comparison.** We compare TRAC with several baselines: full fine-tuning (FT), which updates all model parameters, LoRA (Hu et al., 2022), VeRA (Kopiczko et al., 2024), LoRETTA (Yang et al., 2024), VB-LoRA (Li et al., 2024) and AdaLoRA (Zhang et al., 2023b), all of which were described in the previous section. All experiments were conducted using either the Hugging Face PEFT library (Mangrulkar et al., 2022) or the open-source of the respective methods. Throughout all tables, **bold** and underlined values indicate best and second-best results, respectively.

## 5.1 NATURAL LANGUAGE UNDERSTANDING

### 5.1.1 GLUE BENCHMARKS

We evaluate our method on RoBERTa Base and RoBERTa Large models (Liu, 2019) using the General Language Understanding Evaluation (GLUE) benchmark (Wang et al., 2018) and compare it with other baselines. Following prior work (Kopiczko et al., 2024; Gao et al., 2024; Li et al., 2024), we focus on six GLUE datasets: SST-2 (Socher et al., 2013), MRPC (Dolan & Brockett, 2005), CoLA (Warstadt et al., 2019), QNLI (Wang et al., 2018), RTE (Wang et al., 2018), and STS-B (Cer et al., 2017).

**Implementation Details.** For each task, trainable parameters were initialized independently[2]. As the test labels for GLUE are not publicly available, we report results on the validation set (see Appendix D.1 for details). We compare our method to LoRA, VeRA and LoRETTA. Following LoRA (Hu et al., 2022), we apply all methods only to the query and value projection matrices across all layers while fully fine-tuning the classification head. For all methods, we report the number of trainable parameters attributed to the fine-tuned layers, excluding the classification head, which is fully trained and remains consistent across all methods. We used a batch size of 64 across all experiments, while the number of epochs varied by task. Detailed settings for each method are provided in the Appendix E.1. Each task was run 5 times with different random seeds, using the best epoch's performance for each run, and we report the mean value and standard deviation.

**Results.** Table 3 demonstrates that TRAC consistently outperforms VeRA and LoRETTA across both models while achieving comparable results to LoRA, despite reducing trainable parameters by up to $19\times$ compared to LoRA (0.041M vs. 0.8M).

---

[2]Rather than using the best weights fine-tuned on the MNLI task to initialize MRPC, RTE, and STS-B tasks (Hu et al., 2022).

Table 4: Performance comparison on SuperGLUE tasks with LLaMA2-7B and LLaMA2-13B.

| | METHOD | # TRAIN. PARAM. | CB | BOOLQ | WSC | COPA | AVG. |
|---|---|---|---|---|---|---|---|
| 7B | LoRA | 2.10M | **96.4** $_{\pm1.5}$ | **85.3** $_{\pm1.4}$ | **70.8** $_{\pm1.2}$ | 95.0 $_{\pm0.8}$ | **87.0** |
| | VeRA | 0.279M | 94.0 $_{\pm0.8}$ | 84.5 $_{\pm0.6}$ | 65.7 $_{\pm0.5}$ | 95.0 $_{\pm0.0}$ | 84.8 |
| | LoRETTA | 0.315M | 95.8 $_{\pm1.7}$ | 83.9 $_{\pm0.8}$ | 67.0 $_{\pm2.5}$ | **96.0** $_{\pm2.4}$ | 85.7 |
| | **TRAC** | **0.106M** | 95.8 $_{\pm0.8}$ | 85.0 $_{\pm0.6}$ | 68.3 $_{\pm2.7}$ | 95.3 $_{\pm0.5}$ | 86.1 |
| 13B | LoRA | 3.28M | 97.0 $_{\pm0.8}$ | **87.0** $_{\pm0.6}$ | 73.4 $_{\pm3.0}$ | 95.7 $_{\pm0.5}$ | 88.3 |
| | VeRA | 0.430M | 95.2 $_{\pm1.7}$ | 85.7 $_{\pm0.7}$ | 73.7 $_{\pm1.2}$ | 94.7 $_{\pm1.2}$ | 87.3 |
| | LoRETTA | 0.570M | 94.0 $_{\pm2.2}$ | 86.4 $_{\pm0.7}$ | 73.7 $_{\pm3.9}$ | 96.3 $_{\pm0.5}$ | 87.6 |
| | **TRAC** | **0.166M** | **98.2** $_{\pm0.0}$ | **86.4** $_{\pm1.1}$ | **74.0** $_{\pm3.1}$ | 95.7 $_{\pm1.2}$ | **88.6** |

Table 5: Performance of fine-tuning methods on GPT-2 Medium and Large for the E2E benchmark. Higher values indicate better performance. Results marked with ([1,2,3]) are from previous work: [1](Hu et al., 2022), [2](Zi et al., 2023), [3](Kopiczko et al., 2024), [4](Li et al., 2024).

| | METHOD | # TRAIN. PARAM. | BLEU | NIST | METEOR | ROUGE-L | CIDER |
|---|---|---|---|---|---|---|---|
| MEDIUM | FT[1] | 354.92M | 68.2 | 8.62 | 46.2 | 71.0 | 2.47 |
| | ADPT[L1] | 0.37M | 66.3 | 8.41 | 45.0 | 69.8 | 2.40 |
| | LoRA | 0.35M | 68.9 | 8.69 | 46.5 | 71.3 | 2.51 |
| | AdaLoRA[2] | 0.38M | 68.2 | 8.58 | 44.1 | 70.7 | 2.35 |
| | VeRA[3] | 0.098M | 70.1 | **8.81** | 46.6 | 71.5 | 2.50 |
| | VB-LoRA[4] | 0.076M | 70.0 | **8.81** | 46.6 | 71.5 | 2.52 |
| | **TRAC** | **0.054M** | **70.3** | **8.81** | **46.9** | **71.7** | **2.54** |
| LARGE | FT[1] | 774.03M | 68.5 | 8.78 | 46.0 | 69.9 | 2.45 |
| | ADPT[L1] | 0.88M | 69.1 | 8.68 | 46.3 | 71.4 | 2.49 |
| | LoRA | 0.77M | 70.5 | **8.88** | 46.7 | **72.2** | 2.53 |
| | VeRA[3] | 0.17M | 70.3 | 8.85 | **46.9** | 71.6 | **2.54** |
| | VB-LoRA[4] | 0.13M | 70.3 | 8.86 | 46.7 | **72.2** | 2.54 |
| | **TRAC** | **0.099M** | **70.6** | 8.87 | 46.8 | 71.8 | **2.54** |

### 5.1.2 SUPERGLUE BENCHMARKS

We evaluate our method on LLaMA2-7B and LLaMA2-13B (Touvron et al., 2023) using four SuperGLUE classification tasks—CB, BoolQ, WSC, and COPA (Wang et al., 2019).

**Implementation Details.** Since test labels are not publicly available in SuperGLUE, we report results on the validation set. For large datasets, we sample both training and validation sets. See Appendix D.3 for detailed setup. Following (Malladi et al., 2023), we reformulate all SuperGLUE classification tasks as language modelling problems. Both methods are trained for 10 epochs per task, with results reported from the final epoch. The LoRA matrix rank is set to 4 for both models. The tensor structure of TRAC and additional hyperparameters are detailed in the Appendix E.2.

**Results.** Table 4 demonstrates that for LLaMA2-7B, TRAC outperforms both VeRA and LoRETTA while maintaining comparable performance to LoRA—despite using 20× fewer parameters. For LLaMA2-13B, TRAC achieves the best overall performance among all methods.

### 5.2 NATURAL LANGUAGE GENERATION

We evaluate our method's performance on generative tasks using the E2E benchmark (Novikova et al., 2017) with GPT-2 Medium and Large models (Radford et al., 2019).

**Implementation Details.** Following (Hu et al., 2022), we fine-tune the query and key projection matrices for 5 epochs in both LoRA and our method, reporting results from the final epoch. We adopt the hyperparameters from (Hu et al., 2022) for LoRA implementations, and employ a linear learning rate scheduler to determine the optimal learning rate for ours. Complete hyperparameter settings and tensor structure details are provided in the Appendix E.3.

Table 6: Commonsense reasoning results of LoRA and TRAC on LLaMA3-8B and Qwen3-8B.

| MODEL | METHOD | # TRAIN. PARAM. | BQ | PQ | SQ | HS | WG | AE | AC | OB | AVG. |
|---|---|---|---|---|---|---|---|---|---|---|---|
| LLaMA3-8B | LoRA | 1.70M | 69.39 | 85.31 | 76.00 | 90.77 | 77.27 | **89.44** | **76.37** | **79.00** | 80.44 |
| | TRAC | **0.66M** | 70.52 | **85.69** | 75.90 | 91.29 | **81.45** | 87.84 | 76.28 | 78.40 | 80.92 |
| | TRAC | 1.00M | **71.77** | **85.69** | **77.02** | **91.81** | 81.29 | 88.05 | 76.02 | **79.00** | **81.33** |
| Qwen3-8B | LoRA | 1.92M | 67.31 | **88.79** | 79.38 | 90.26 | 76.16 | **96.38** | **90.87** | **89.40** | 84.82 |
| | TRAC | **0.67M** | 66.67 | 87.49 | 79.63 | 92.16 | 80.27 | 96.17 | 90.44 | 88.40 | 85.15 |
| | TRAC | 1.06M | **70.95** | 88.74 | **79.99** | **92.37** | **80.98** | 96.34 | 90.61 | 89.20 | **86.15** |

Table 7: Results of LoRA and TRAC on GSM8K accuracy and HumanEval pass@1 using the LLaMA3-8B model.

| METHOD | # TRAIN. PARAM. | GSM8K | HUMANEVAL |
|---|---|---|---|
| LoRA | 1.70M | 64.67 | 38.78 |
| **TRAC** | **0.66M** | 63.31 | 38.29 |
| **TRAC** | **1.00M** | **64.90** | **38.90** |

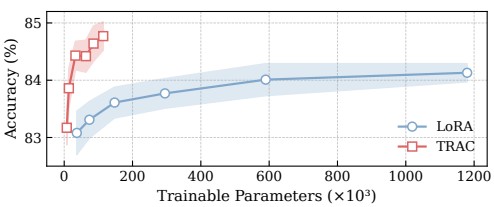

Figure 2: Accuracy vs. Parameter Count of LoRA and TRAC on ViT-Base (CIFAR-100).

**Results.** As shown in Table 5, TRAC uses only 14% and 12% of LoRA's trainable parameters on GPT-2 Medium and Large, and about half of VeRA's. Despite using nearly one order of magnitude fewer parameters than AdaLoRA, TRAC achieves higher scores. Compared with VB-LoRA—which reports stored rather than trainable parameters (see Appendix C.2)—TRAC attains comparable or better performance with greater practical efficiency. Overall, it matches or surpasses all baselines across most metrics, demonstrating strong performance under a drastically reduced parameter budget.

## 5.3 COMMONSENSE REASONING, MATH, AND CODE BENCHMARKS

We further evaluate our method on commonsense reasoning, mathematical reasoning, and code generation tasks with LLaMA3-8B (AI@Meta, 2024) and Qwen3-8B (Team, 2025) models.

**Implementation Details.** We follow the experimental settings of DoRA (Liu et al., 2024a) and LoRA-GA (Wang et al., 2024). For commonsense reasoning, the models are trained on Commonsense170K and evaluated on eight standard benchmarks (Hu et al., 2023). For mathematical reasoning, the models are trained on a 100k subset of MetaMathQA (Yu et al., 2023) and evaluated on GSM8K (Cobbe et al., 2021). For code generation, the models are trained on a 100k subset of Code-Feedback (Zheng et al., 2024) and evaluated on HumanEval (Chen, 2021). All experiments are conducted for 1 epoch, with performance evaluated at the end of training. For both models, the LoRA matrix rank is set to 4, while TRAC is tested under two parameter configurations. Detailed hyperparameter and configuration settings are available in the Appendix E.4.

**Results.** Tables 6–7 summarize the results across the three benchmarks. Across all tasks, TRAC achieves comparable or superior performance to LoRA while using only 35%–59% of its trainable parameters, demonstrating efficiency and robust generalization across modern models and tasks.

## 5.4 IMAGE CLASSIFICATION

We evaluate our method on vision tasks using ViT Base and ViT Large models (Dosovitskiy et al., 2020; Wu et al., 2020) pretrained on ImageNet-21k (Deng et al., 2009). The evaluation is conducted on four image classification benchmarks: CIFAR-10/100 (Alex, 2009), Flowers102 (Nilsback & Zisserman, 2008), and CUB-200-2011 (Welinder et al., 2010).

**Implementation Details.** Following (Koohpayegani et al., 2024), we use 10 labeled samples per class to simulate a low-data regime (Appendix D.4). We compare TRAC with full fine-tuning (FT), classification head tuning (Head), LoRA, and LoRETTA. All methods update the final classification layer (excluded from parameter counts); PEFT methods also tune the query and key projections, with LoRA's rank set to 8 for both ViT-Base and ViT-Large. TRAC's tensor structure and other hyperparameters are given in Appendix E.5. Models are trained for 50 epochs with a batch size of 16. We use four data-sampling and three initialization seeds (12 runs total) and report mean top-1 accuracy and standard deviation from the final epoch.

Table 8: Accuracy of fine-tuned ViT Base and Large models on image classification benchmarks.

| | METHOD | # TRAIN. PARAM. | CIFAR-10 | CIFAR-100 | FLOWERS102 | CUB-200-2011 | AVG. |
|---|---|---|---|---|---|---|---|
| BASE | FULL | 85.840M | $87.03_{\pm1.70}$ | $76.49_{\pm1.71}$ | $99.04_{\pm0.08}$ | $82.07_{\pm0.44}$ | 86.16 |
| | HEAD | - | $87.60_{\pm0.44}$ | $67.45_{\pm0.52}$ | $98.60_{\pm0.04}$ | $81.20_{\pm0.15}$ | 83.71 |
| | LoRA | 0.295M | $\underline{93.60}_{\pm0.40}$ | $\underline{83.84}_{\pm0.27}$ | $\mathbf{99.38}_{\pm0.04}$ | $84.25_{\pm0.24}$ | $\underline{90.27}$ |
| | LoRETTA | 0.068M | $93.02_{\pm0.40}$ | $83.21_{\pm0.21}$ | $99.29_{\pm0.06}$ | $\underline{84.75}_{\pm0.40}$ | 90.07 |
| | **TRAC** | **0.026M** | $\mathbf{94.86}_{\pm0.38}$ | $\mathbf{84.21}_{\pm0.40}$ | $\underline{99.30}_{\pm0.07}$ | $\mathbf{85.04}_{\pm0.34}$ | **90.85** |
| LARGE | FULL | 304.088M | $93.91_{\pm0.90}$ | $82.52_{\pm0.91}$ | $99.25_{\pm0.06}$ | $82.87_{\pm0.25}$ | 89.64 |
| | HEAD | - | $92.86_{\pm0.25}$ | $75.71_{\pm0.42}$ | $98.88_{\pm0.04}$ | $83.03_{\pm0.17}$ | 87.62 |
| | LoRA | 0.786M | $95.87_{\pm0.45}$ | $\underline{86.50}_{\pm0.20}$ | $\mathbf{99.54}_{\pm0.06}$ | $85.05_{\pm0.16}$ | 91.74 |
| | LoRETTA | 0.154M | $\underline{96.24}_{\pm0.39}$ | $86.31_{\pm0.36}$ | $\underline{99.49}_{\pm0.04}$ | $86.42_{\pm0.30}$ | $\underline{92.12}$ |
| | **TRAC** | **0.056M** | $\mathbf{96.81}_{\pm0.25}$ | $\mathbf{87.10}_{\pm0.28}$ | $\underline{99.49}_{\pm0.05}$ | $\mathbf{86.99}_{\pm0.35}$ | **92.60** |

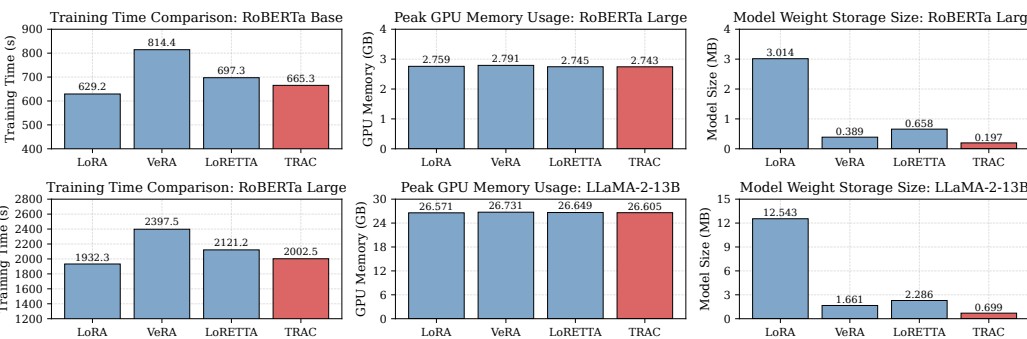

Figure 3: Comparison of TRAC with main baselines on the RoBERTa and LLaMA-2-13B models, covering (a) training time, (b) peak GPU memory usage, and (c) model weight storage size.

**Results.** Table 8 demonstrates that TRAC outperforms LoRA on three datasets with $11\times$ fewer parameters for ViT Base and maintains comparable performance with $14\times$ fewer parameters for ViT Large. Additionally, TRAC consistently surpasses LoRETTA across all configurations while using only 1/3 of its parameter count.

## 5.5 PARAMETER SENSITIVITY AND EFFICIENCY ANALYSIS

**Parameter Sensitivity Analysis.** To evaluate how performance varies with parameter size, we conducted experiments on the ViT-Base model by adjusting the number of trainable parameters for both TRAC and LoRA. As shown in Figure 2, performance for both methods improves steadily as parameter capacity increases. However, TRAC consistently achieves higher accuracy with significantly fewer parameters, highlighting its more favorable parameter–efficiency trade-off.

**Time and Memory Efficiency Evaluation.** We conducted additional quantitative comparisons of training time, GPU memory usage, and storage cost relative to the main baselines. As shown in Figure 3, the additional training-time overhead of TRAC remains below 6% relative to LoRA, indicating that the computational cost introduced by tensor decomposition is negligible. Meanwhile, the peak GPU memory consumption during training is almost identical across methods, and TRAC reduces model weight storage by more than 93% relative to LoRA. Overall, these results show that TRAC achieves strong parameter compression with minimal computational overhead.

## 6 CONCLUSION

In this paper, we propose a novel fine-tuning method that combines parameter freezing and sharing with Tensor-Train decomposition, significantly reducing trainable parameters while retaining LoRA's advantages. Across various language and vision models and benchmarks, our method achieves comparable or superior performance to LoRA and its variants while significantly reducing parameter counts. Several promising research directions emerge from this work. Theoretically, rigorous analysis is needed to understand how parameter freezing and sharing affect the representational capacity of Tensor-Train decomposition. Practically, developing efficient strategies to search for optimal configurations of tensor structures and ranks remains an important direction for future research.

ETHICS STATEMENT

We confirm adherence to the ICLR Code of Ethics. This work proposes a general-purpose parameter-efficient fine-tuning algorithm (TRAC) using only publicly available datasets and open-source models, involving no human subjects or personally identifiable information. While our method improves computational efficiency, we acknowledge that it facilitates model adaptation, potentially including malicious use cases. However, TRAC is content-agnostic; any ethical risks (e.g., bias, toxicity) stem from the base models or downstream data. Responsibility for safe deployment lies with the practitioners. The authors declare no conflicts of interest.

REPRODUCIBILITY STATEMENT

We ensure the reproducibility of our work through detailed documentation of theoretical analysis, experimental setups, and open-source code. The derivations and discussions supporting our approximation analysis and complexity comparisons (Section 4) are provided in Appendix B. The experimental settings are described in Section 5, with a comprehensive list of hyperparameters for all evaluated architectures (e.g., Qwen, LLaMA, GPT, BERT, and ViT) detailed in Appendix E. Our implementation of TRAC is available at `https://github.com/BangguoYe/TRAC`.

ACKNOWLEDGMENTS

The authors would like to thank the anonymous reviewers for their valuable comments. This work was supported by the National Natural Science Foundation of China (Grant No. 12090024 and Grant No. 125B2026) and the Natural Science Foundation of Chongqing, China (Grant No. CSTB2023NSCQ-LZX0054).

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

# A  SUPPLEMENTARY DETAILS OF TRAC

*Note:* For all ablation experiments in this section, we fix the lightweight vector controller's activation to a linear function $\sigma(x) = x$, in order to control for the effect of activation choices. This setting differs slightly from the main experimental section, where task-dependent activations are used.

## A.1  INITIALIZATION STRATEGIES

**Tensor-Train Uniform Initialization.** Based on the parameter $\sigma_k$ defined in (7), we can consider another option that uses the uniform distribution within the range $[-\sigma_k, \sigma_k]$, and we call this approach the Tensor-Train uniform initialization.

We use two types of Kaiming initialization: normal distribution and uniform distribution.

**Kaiming Normal Initialization (He et al., 2015).** Parameters are initialized following a normal distribution $\mathcal{N}(0, \sigma_h^2)$, where the variance $\sigma_h^2$ is defined as

$$\sigma_h^2 = \frac{2}{n_{in}}, \tag{8}$$

where $n_{in}$ denotes the dimension of the input features. In LoRA, we apply variances of $\frac{2}{n}$ and $\frac{2}{r}$ for $A^T \in \mathbb{R}^{r \times n}$ and $B \in \mathbb{R}^{m \times r}$, respectively.

**Kaiming Uniform Initialization (He et al., 2015).** For uniform distribution, the initialization ranges are $[-\sqrt{\frac{2}{n}}, \sqrt{\frac{2}{n}}]$ and $[-\sqrt{\frac{2}{r}}, \sqrt{\frac{2}{r}}]$ for $A^T \in \mathbb{R}^{r \times n}$ and $B \in \mathbb{R}^{m \times r}$, respectively.

We evaluate the impact of different initialization approaches on TRAC performance. Four initialization strategies were tested:

- **TT Norm.**: Tensor-Train normal initialization using the formula defined in Equation 7, which accounts for tensor ranks and shape factors.

- **TT Unif.**: A uniform variant of Tensor-Train initialization, where values are sampled from a uniform distribution scaled by factors similar to those in TT Norm.

- **Kaiming Norm.**: Standard Kaiming normal initialization (He et al., 2015), commonly used in deep neural networks.

- **Kaiming Unif.**: Kaiming uniform initialization (He et al., 2015), providing a uniform distribution alternative to Kaiming normal.

As demonstrated in Table 9, the Tensor-Train normal initialization consistently outperforms other strategies across all three evaluation tasks. This confirms the importance of properly accounting for tensor structure during initialization to ensure stable training and optimal performance in tensor-based adaptation methods.

Table 9: Ablation experiments comparing different initialization strategies on MRPC, CoLA, and RTE datasets. The corresponding model is RoBERTa Base.

| INIT. STRATEGY | MRPC | CoLA | RTE |
|---|---|---|---|
| TT NORM. | **89.0** $_{\pm0.7}$ | **65.2** $_{\pm2.4}$ | **79.9** $_{\pm3.6}$ |
| TT UNIF. | 87.7 $_{\pm1.1}$ | 62.4 $_{\pm1.5}$ | 77.7 $_{\pm2.2}$ |
| KAINMING NORM. | 87.9 $_{\pm1.3}$ | 63.8 $_{\pm2.6}$ | 79.6 $_{\pm2.0}$ |
| KAINMING UNIF. | 87.8 $_{\pm0.5}$ | 63.3 $_{\pm2.6}$ | **79.9** $_{\pm1.5}$ |

Table 10: Ablation experiments comparing TRAC with its variants on CoLA and CIFAR-100 datasets. The corresponding models are RoBERTa Base and ViT Base.

| # TRAIN. MODULE | # TRAIN. PARAM. | CoLA | CIFAR-100 |
|---|---|---|---|
| TRAC (OURS) | 0.026M | 65.2 | 84.21 |
| SHARED TT | 0.025M | 61.63 ($\downarrow$ 5.4 %) | 84.14 ($\downarrow$ 0.1%) |
| FROZEN | 0.008M | 38.98 ($\downarrow$ 40 %) | 67.47 ($\downarrow$ 19 %) |
| ALL TRAINABLE | 0.41M | 66.17 ($\uparrow$ 1.6 %) | 84.44 ($\uparrow$ 0.3%) |

## A.2 COMPONENT AND PARAMETER ANALYSIS

We systematically investigate the impact of tensor and vector components in our method, and further analyze the parameter efficiency of different designs.

**Component Study.** We compare four configurations:

- **TRAC (Ours)**: The complete TRAC method as described in Section 3.4.
- **Shared TT**: Removal of all vector controllers, keeping only shared tensor cores.
- **Frozen**: Replacing trainable tensors with frozen tensors to assess the impact of adaptability.
- **All Trainable**: Converting all frozen and shared tensors to independent trainable parameters.

As shown in Table 10, both shared tensor cores and vector controllers make substantial contributions to model performance. Removing controllers (Shared TT) or using frozen tensors (Frozen) leads to notable performance degradation. While training all tensor cores independently (All Trainable) provides marginal performance gains, it requires drastically more parameters, making it significantly less parameter-efficient. These results validate our design decisions, confirming that TRAC achieves an effective balance between performance and parameter efficiency.

**Parameter Efficiency Analysis.** We now provide a detailed analysis of the trainable parameter counts for the three approximation methods (All Trainable, Shared TT, and TRAC), complementing the performance study above.

**All Trainable:** For each adaptation matrix $A^{(i)} \in \mathbb{R}^{n \times r}$, this method uses three independent trainable tensors:
$$\mathcal{G}_1^{(i)} \in \mathbb{R}^{1 \times n_1 \times r_1}, \quad \mathcal{G}_2^{(i)} \in \mathbb{R}^{r_1 \times n_2 \times r_2}, \quad \mathcal{G}_3^{(i)} \in \mathbb{R}^{r_2 \times n_3 \times r}.$$

The parameter count for each component is:

- $\mathcal{G}_1^{(i)}$: $1 \times n_1 \times r_1 = n_1 r_1$ parameters
- $\mathcal{G}_2^{(i)}$: $r_1 \times n_2 \times r_2 = n_2 r_1 r_2$ parameters
- $\mathcal{G}_3^{(i)}$: $r_2 \times n_3 \times r = n_3 r_2 r$ parameters

For all $L$ matrices, the total number of trainable parameters is:
$$L(n_1 r_1 + n_2 r_1 r_2 + n_3 r_2 r).$$

**Shared TT:** This method utilizes:

- $L$ independent tensors $\{\mathcal{G}_1^{(i)}\}_{i=1}^L$, each with $n_1 r_1$ parameters
- One frozen tensor $\mathcal{G}_2 \in \mathbb{R}^{r_1 \times n_2 \times (kr_2)}$ (initialized randomly but kept fixed, hence not counted as trainable)
- One shared trainable tensor $\mathcal{G}_3 \in \mathbb{R}^{(kr_2) \times n_3 \times r}$, contributing $kn_3 r_2 r$ parameters

The total number of trainable parameters is:
$$L \times n_1 r_1 + kn_3 r_2 r = Ln_1 r_1 + kn_3 r_2 r.$$

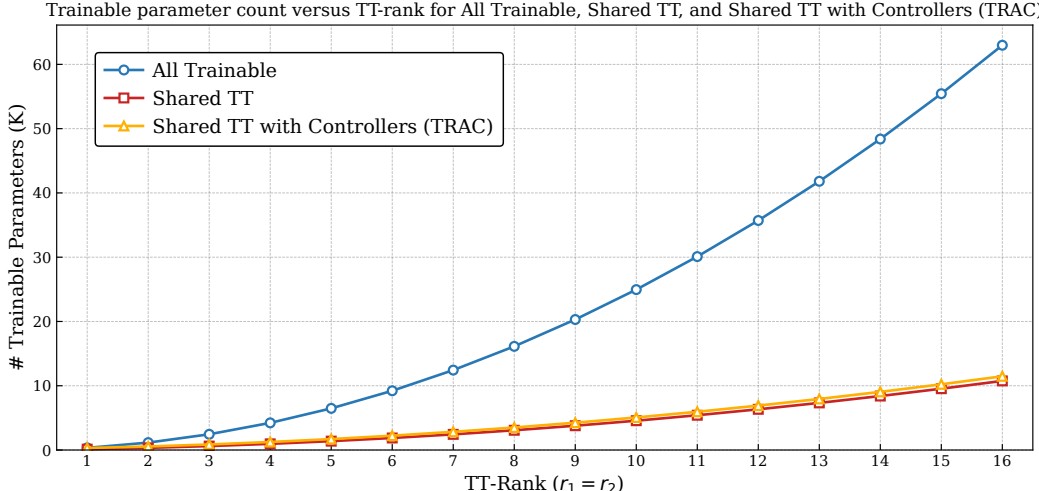

Figure 4: Trainable parameter count versus TT-rank for All Trainable, Shared TT, and Shared TT with Controllers (TRAC) (Set $L = 12$, $n_1 = 8$, $n_2 = 8$, $n_3 = 12$, $k = 3$).

Note that the parameter saving comes primarily from sharing $\mathcal{G}_3$ across all $L$ matrices and using a frozen $\mathcal{G}_2$.

**Shared TT with Controllers (TRAC):** This method enhances the Shared TT by using:

- $L$ tensors $\{\mathcal{G}_1^{(i)}\}_{i=1}^L$: $L \times n_1 r_1$ parameters
- One shared tensor $\mathcal{G}_3$: $k n_3 r_2 r$ parameters
- $L$ controller vectors $\{\mathbf{b}^{(i)}\}_{i=1}^L$: $L \times k r_2$ parameters
- $L$ controller vectors $\{\mathbf{d}^{(i)}\}_{i=1}^L$: $L \times n_3$ parameters

The total number of trainable parameters is:

$$L n_1 r_1 + k n_3 r_2 r + L k r_2 + L n_3 = L(n_1 r_1 + k r_2 + n_3) + k n_3 r_2 r.$$

While this method adds more parameters compared to the Shared TT approach, the additional controllers provide layer-specific customization of the shared components, significantly enhancing expressivity with only a modest increase in parameter count. Specifically, the controllers add only $L(k r_2 + n_3)$ parameters, which is typically much smaller than the $L \times n_2 r_1 r_2$ parameters saved by freezing $\mathcal{G}_2$ and the $(L-1) \times n_3 r_2 r$ parameters saved by sharing $\mathcal{G}_3$, thereby confirming that the additional cost of controllers is negligible compared to the savings.

As illustrated in Figure 4, even though the TT-rank is enlarged by a factor of $k$ in shared methods, both Shared TT and TRAC require far fewer trainable parameters than the All Trainable TT approach across all rank settings. TRAC in particular achieves this efficiency while offering additional flexibility through controllers, making it a highly effective and scalable solution.

## A.3 IMPACT OF PARAMETER ALLOCATION ASYMMETRY

We present additional experiments that demonstrate how asymmetric parameter allocation between matrices $A$ and $B$ affects model performance.

Previous work, such as LoRA-FA (Zhang et al., 2023a), indicated that the $B$ matrix plays a more critical role than the $A$ matrix in LoRA adaptation. While LoRA-FA completely freezes the $A$ matrix, our tensor-based approach enables more fine-grained control over parameter allocation between matrices.

**Symmetry Analysis of Tensor Structure.** Across different models and tasks, matrices $A$ and $B$ in LoRA may exhibit varying levels of importance, which is reflected in the tensor ranks used in our

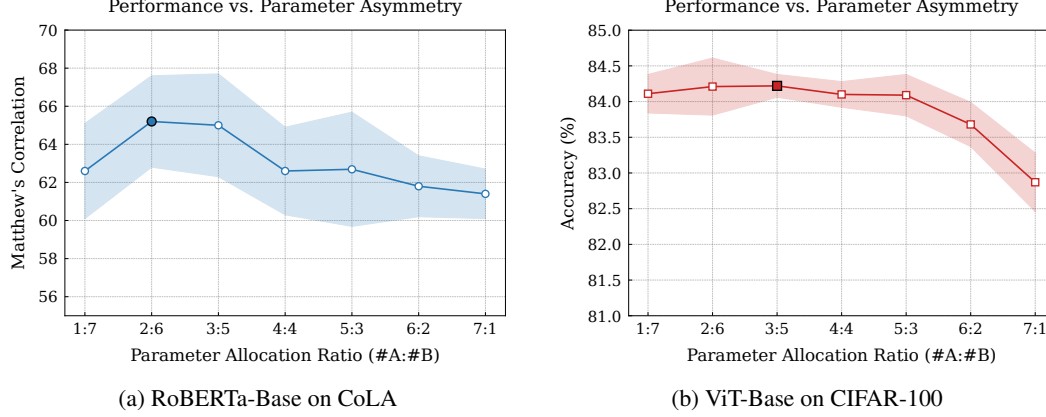

(a) RoBERTa-Base on CoLA      (b) ViT-Base on CIFAR-100

Figure 5: Performance comparison with different parameter allocation ratios between $A$ and $B$ matrices. Both plots demonstrate that asymmetric configurations favoring more parameters in the $B$ matrix (ratios 2:6 and 3:5) yield better performance than symmetric (4:4) or $A$-heavy allocations.

approach. For asymmetric structures, we set tensor ranks $(r_1, r_2)$ for matrix $A$, while larger tensor ranks $(3r_1, 3r_2)$ are allocated to matrix $B$, resulting in a $3\times$ parameter count for $B$ compared to $A$. For symmetric structures, we use tensor ranks $(2r_1, 2r_2)$ for both matrices. Table 11 shows that for various models and tasks, asymmetric tensor structures consistently perform better.

Table 11: Performance comparison of symmetric versus asymmetric tensor structures across CoLA (Matthew's correlation), CIFAR-100 (accuracy), and E2E (BLEU). The corresponding models are RoBERTa-base, ViT-base and GPT2-base.

| TASK | CoLA | CIFAR-100 | E2E |
|---|---|---|---|
| SYMMETRY | $62.6_{\pm 2.3}$ | $84.10_{\pm 0.18}$ | 69.4 |
| ASYMMETRY | $\mathbf{65.2}_{\pm 2.4}$ | $\mathbf{84.21}_{\pm 0.40}$ | $\mathbf{70.3}$ |

To further investigate the optimal degree of asymmetry, we conducted more fine-grained experiments on two representative tasks: RoBERTa-Base on CoLA and ViT-Base on CIFAR-100. In each experiment, we maintained a constant total number of trainable parameters while varying the allocation ratio between matrices $A$ and $B$. Figure 5 visualizes these results.

The experiments reveal a clear pattern: asymmetric configurations that allocate more parameters to the $B$ matrix and fewer to the $A$ matrix (e.g., ratios 2:6 and 3:5) consistently outperform both symmetric allocations (4:4) and configurations that favor the $A$ matrix (e.g., 6:2, 7:1). Specifically, the optimal ratio appears to be 2:6 for RoBERTa-Base on CoLA (65.2 Matthew's correlation) and 3:5 for ViT-Base on CIFAR-100 (84.22% accuracy).

In contrast, when more parameters are allocated to the $A$ matrix (ratios 6:2 and 7:1), we observe noticeable performance degradation. This empirical evidence confirms that:

1. The $B$ matrix requires more trainable parameters than the $A$ matrix, consistent with previous research suggesting its greater importance in LoRA-based adaptation (Zhu et al., 2024).

2. Our approach's ability to flexibly distribute parameters between matrices enables more effective exploitation of this inherent asymmetry.

3. Architectural flexibility provides tangible performance gains by allowing practitioners to optimize parameter allocation based on the specific characteristics of different models and tasks.

These findings validate our claims about the benefits of parameter granularity and architectural flexibility in our proposed method, demonstrating concrete performance improvements through tailored parameter allocation. The results are consistent across multiple models (RoBERTa, ViT,

GPT2) and tasks (CoLA, CIFAR-100, E2E), indicating that the asymmetric nature of LoRA adaptation is a general phenomenon that our tensor-based approach effectively leverages.

## A.4    HYPERPARAMETER SENSITIVITY AND RECOMMENDED CHOICES

We provide additional details on hyperparameter selection, which were also clarified during the rebuttal stage.

**Tensor order.** A 3rd-order tensor suffices to capture the design of trainable, frozen, and shared cores. If the tensor order is increased to $d > 3$, then in TT decomposition most middle cores have shape $r \times n_i \times r$ for $i = 2, \ldots, d-1$, with parameter count $O(r^2)$. To maintain a total budget of $O(r)$, these middle cores must remain frozen, leaving little benefit while increasing computational cost. We compare 3rd-, 4th- and 5th-order settings under the same trainable budget in Table 12, confirming that higher orders degrade efficiency without improving performance.

Table 12: Comparison of different tensor orders under the same trainable parameter budget. Performance is reported on the RoBERTa-Base setting.

| Tensor Order | Train. Param. | CoLA | MRPC | RTE | STS-B |
|---|---|---|---|---|---|
| **3RD-ORDER** | **26,304** | **64.8**$_{\pm 2.9}$ | **88.6**$_{\pm 1.8}$ | **80.1**$_{\pm 1.8}$ | **91.2**$_{\pm 0.2}$ |
| 4TH-ORDER | 26,784 | 64.5$_{\pm 1.2}$ | 87.5$_{\pm 1.4}$ | 75.3$_{\pm 2.8}$ | 90.8$_{\pm 0.2}$ |
| 5TH-ORDER | 28,560 | 63.7$_{\pm 1.5}$ | **88.6**$_{\pm 0.6}$ | 76.4$_{\pm 3.1}$ | 90.7$_{\pm 0.2}$ |

Training efficiency results are provided in Table 13. As shown, higher-order tensors require longer training time, leading to slightly reduced efficiency.

Table 13: Training time per epoch with different tensor orders.

| Tensor Order | 3rd-order | 4th-order | 5th-order |
|---|---|---|---|
| TRAINING TIME / S | 776.3 | 812.5 | 873.7 |

**Tensor ranks.** The number of trainable parameters can be written as

$$\#p \approx L(n_1 r_1 + k r_2 + n_3) + k n_3 r_2 r,$$

where $L$ is the number of layers, $r$ the LoRA rank, and $k$ a scaling factor related to controller vectors. This allows us to directly solve for $(r_1, r_2)$ given a target budget $\#p$. In practice, setting $r_1 = r_2$ allows us to directly solve for their values under a given parameter budget, providing a convenient starting point. Subsequently, the asymmetry property (cf. Section 3.5) can further reallocate ranks between matrices $A$ and $B$, typically assigning larger ranks to $B$ when beneficial.

In conclusion, these results confirm that (i) increasing tensor order beyond 3rd is unnecessary, and (ii) tensor ranks can be systematically set according to parameter budgets, rather than arbitrary tuning.

# B  THEORETICAL ANALYSIS AND DETAILS

## B.1  DERIVATION OF THE ERROR BOUND FOR TT-SVD

In this section, we present the detailed derivation of the observations discussed in Section 4.1.

First, we analyze the approximation error under given TT ranks. Note that while the general error analysis of TT-SVD is established in (Oseledets, 2011), we specialize the analysis here to the exact tensorized structure arising in our framework. This concrete instance explicates the roles of the particular tensor shapes and unfoldings involved in TRAC.

**Setup Recall.** We briefly recall the relevant structure introduced in Section 3. The weight matrix $A \in \mathbb{R}^{n \times r}$ is reshaped into a fourth-order tensor

$$\mathcal{A} = \text{Reshape}(A, (n_1, n_2, n_3), r) \ \in \ \mathbb{R}^{n_1 \times n_2 \times n_3 \times r}, \qquad n_1 n_2 n_3 = n.$$

Its TT approximation $\hat{\mathcal{A}}$ follows the tensor-train format

$$\hat{\mathcal{A}}(i_1, i_2, i_3, :) = \mathcal{G}_1(i_1)\,\mathcal{G}_2(i_2)\,\mathcal{G}_3(i_3),$$

with TT-rank $(1, r_1, r_2)$ and cores $\mathcal{G}_1 \in \mathbb{R}^{1 \times n_1 \times r_1}$, $\mathcal{G}_2 \in \mathbb{R}^{r_1 \times n_2 \times r_2}$, $\mathcal{G}_3 \in \mathbb{R}^{r_2 \times n_3 \times r}$. The derivation below characterizes the error $\|\mathcal{A} - \hat{\mathcal{A}}\|_F$ when these cores are obtained via TT-SVD.

**Specialized TT-SVD Error Analysis.** Let $\hat{\mathcal{A}}$ be the TT approximation of $\mathcal{A} \in \mathbb{R}^{n_1 \times n_2 \times n_3 \times r}$ produced by TT-SVD with intermediate ranks $r_1$ and $r_2$, as described above. Let $A_{(1)} \in \mathbb{R}^{n_1 \times (n_2 n_3 r)}$ denote the first-mode unfolding of $\mathcal{A}$, and let $A_{(2)} \in \mathbb{R}^{(r_1 n_2) \times (n_3 r)}$ denote the reshaped coefficient matrix arising after the first truncation step; their precise definitions are given in Steps 1–2 below (note in particular that $A_{(2)}$ depends on $r_1$). The approximation error satisfies:

$$\|\mathcal{A} - \hat{\mathcal{A}}\|_F^2 = \sum_{k=r_1+1}^{\min(n_1,\, n_2 n_3 r)} \sigma_k(A_{(1)})^2 + \sum_{k=r_2+1}^{\min(r_1 n_2,\, n_3 r)} \sigma_k(A_{(2)})^2,$$

where $\sigma_k(\cdot)$ denotes the $k$-th largest singular value. Consequently, exact recovery is achievable when $r_1 \geq \text{rank}(A_{(1)})$ and $r_2 \geq \text{rank}(A_{(2)})$.

**Derivation Details.** The result is derived by two successive SVD truncations on the unfoldings $A_{(1)}$ and $A_{(2)}$, invoking the classical Eckart–Young–Mirsky theorem. For completeness, we provide the step-by-step procedure below.

**Lemma B.1** (Eckart–Young–Mirsky). *Let $M \in \mathbb{R}^{p \times q}$ with singular values $\sigma_1(M) \geq \sigma_2(M) \geq \cdots \geq 0$. For any $0 \leq s \leq \min(p, q)$, let $M^{(s)}$ denote the rank-$s$ approximation obtained by retaining the $s$ largest singular values. Then*

$$\|M - M^{(s)}\|_F^2 = \sum_{k=s+1}^{\min(p,q)} \sigma_k(M)^2.$$

This lemma asserts the optimality of SVD in the Frobenius norm (Eckart & Young, 1936; Mirsky, 1960; Golub & Van Loan, 2013).

**Step 1: First-Mode Unfolding and Truncation.** Form the first-mode unfolding

$$A_{(1)}\big[i_1,\ (i_2-1)n_3 r + (i_3-1)r + j\big] = \mathcal{A}(i_1, i_2, i_3, j),$$

giving $A_{(1)} \in \mathbb{R}^{n_1 \times (n_2 n_3 r)}$ with full SVD $A_{(1)} = U_1 \Sigma_1 V_1^\top$. Let $\widetilde{U}_1 \in \mathbb{R}^{n_1 \times r_1}$, $\widetilde{\Sigma}_1 \in \mathbb{R}^{r_1 \times r_1}$, and $\widetilde{V}_1 \in \mathbb{R}^{(n_2 n_3 r) \times r_1}$ denote the leading $r_1$ SVD factors. Define the rank-$r_1$ approximation and the residual coefficient matrix:

$$A_{(1)}^{(r_1)} = \widetilde{U}_1 \widetilde{\Sigma}_1 \widetilde{V}_1^\top, \qquad C_1 = \widetilde{\Sigma}_1 \widetilde{V}_1^\top \in \mathbb{R}^{r_1 \times (n_2 n_3 r)}.$$

Set the first TT-core $\mathcal{G}_1(1, i_1, k_1) = \widetilde{U}_1(i_1, k_1)$. By the Eckart–Young–Mirsky lemma,

$$\big\|A_{(1)} - A_{(1)}^{(r_1)}\big\|_F^2 = \sum_{k=r_1+1}^{\min(n_1,\, n_2 n_3 r)} \sigma_k(A_{(1)})^2. \tag{9}$$

Denote by $\hat{\mathcal{A}}_1$ the intermediate tensor whose first-mode unfolding equals $A_{(1)}^{(r_1)} = \widetilde{U}_1 C_1$; then $\|\mathcal{A} - \hat{\mathcal{A}}_1\|_F^2$ equals the right-hand side of (9).

**Step 2: Second-Mode Unfolding and Truncation.** Reshape $C_1 \in \mathbb{R}^{r_1 \times (n_2 n_3 r)}$ into $A_{(2)} \in \mathbb{R}^{(r_1 n_2) \times (n_3 r)}$ via

$$A_{(2)}\big[(k_1 - 1)n_2 + i_2, \ (i_3 - 1)r + j\big] = C_1\big[k_1, \ (i_2 - 1)n_3 r + (i_3 - 1)r + j\big].$$

This operation is a bijective re-indexing of entries, so the Frobenius norm is preserved: $\|A_{(2)}\|_F = \|C_1\|_F$. Perform the SVD $A_{(2)} = U_2 \Sigma_2 V_2^\top$, with leading $r_2$ factors $\widetilde{U}_2 \in \mathbb{R}^{(r_1 n_2) \times r_2}$, $\widetilde{\Sigma}_2 \in \mathbb{R}^{r_2 \times r_2}$, $\widetilde{V}_2 \in \mathbb{R}^{(n_3 r) \times r_2}$. Define:

$$A_{(2)}^{(r_2)} = \widetilde{U}_2 \, \widetilde{\Sigma}_2 \, \widetilde{V}_2^\top, \qquad C_2 = \widetilde{\Sigma}_2 \, \widetilde{V}_2^\top \in \mathbb{R}^{r_2 \times (n_3 r)}.$$

Set the second TT-core $\mathcal{G}_2(k_1, i_2, k_2) = \widetilde{U}_2((k_1 - 1)n_2 + i_2, \ k_2)$, and let $\hat{C}_1 \in \mathbb{R}^{r_1 \times (n_2 n_3 r)}$ be the matrix obtained from $A_{(2)}^{(r_2)}$ by the inverse of the above re-indexing. Since the reshape is an isometry and by the Eckart–Young–Mirsky lemma,

$$\|C_1 - \hat{C}_1\|_F^2 = \big\|A_{(2)} - A_{(2)}^{(r_2)}\big\|_F^2 = \sum_{k=r_2+1}^{\min(r_1 n_2, \, n_3 r)} \sigma_k(A_{(2)})^2. \tag{10}$$

**Step 3: Third Core.** Reshape $C_2$ into the third TT-core: $\mathcal{G}_3(k_2, i_3, j) = C_2[k_2, (i_3 - 1)r + j]$, giving $\mathcal{G}_3 \in \mathbb{R}^{r_2 \times n_3 \times r}$. No further truncation is applied, so no additional error is introduced.

**Combining the Two Error Terms.** Let $E_1 = \mathcal{A} - \hat{\mathcal{A}}_1$ and $E_2 = \hat{\mathcal{A}}_1 - \hat{\mathcal{A}}$, so that $\mathcal{A} - \hat{\mathcal{A}} = E_1 + E_2$. Their first-mode unfoldings are

$$\mathrm{unfold}_1(E_1) = A_{(1)} - A_{(1)}^{(r_1)} = \sum_{k > r_1} \sigma_k(A_{(1)}) \, u_k v_k^\top,$$

$$\mathrm{unfold}_1(E_2) = \widetilde{U}_1 \, (C_1 - \hat{C}_1),$$

where $u_k$ and $v_k$ are the $k$-th left and right singular vectors of $A_{(1)}$. The column space of $\mathrm{unfold}_1(E_1)$ is $\mathrm{span}\{u_k : k > r_1\}$, while the column space of $\mathrm{unfold}_1(E_2)$ is contained in $\mathrm{range}(\widetilde{U}_1) = \mathrm{span}\{u_k : k \leq r_1\}$. These subspaces are mutually orthogonal by the orthonormality of the left singular vectors of $A_{(1)}$, so

$$\langle E_1, \ E_2 \rangle_F = \big\langle \mathrm{unfold}_1(E_1), \ \mathrm{unfold}_1(E_2) \big\rangle_F = 0.$$

By the Pythagorean theorem for the Frobenius norm:

$$\|\mathcal{A} - \hat{\mathcal{A}}\|_F^2 = \|E_1\|_F^2 + \|E_2\|_F^2. \tag{11}$$

For $\|E_2\|_F^2$, note that left-multiplication by a matrix with orthonormal columns is an isometry, so

$$\|E_2\|_F^2 = \|\widetilde{U}_1(C_1 - \hat{C}_1)\|_F^2 = \|C_1 - \hat{C}_1\|_F^2.$$

Substituting (9) and (10) into (11) yields:

$$\|\mathcal{A} - \hat{\mathcal{A}}\|_F^2 = \sum_{k=r_1+1}^{\min(n_1, \, n_2 n_3 r)} \sigma_k(A_{(1)})^2 + \sum_{k=r_2+1}^{\min(r_1 n_2, \, n_3 r)} \sigma_k(A_{(2)})^2.$$

This shows that the total approximation error equals exactly the sum of the squared tail singular values of the two successive unfoldings. Exact recovery follows when both sums vanish, which holds if and only if $r_1 \geq \mathrm{rank}(A_{(1)})$ and $r_2 \geq \mathrm{rank}(A_{(2)})$.

## B.2 Analysis of Error Bound for Shared Structures

We now extend the analysis to Tensor-Train structures shared across layers within a specific family of matrices.

**Shared-core approximation for structured matrix families.** Consider a family of matrices $\{A^l\}_{l=1}^L \subset \mathbb{R}^{n \times r}$. Assume that each $A^l$ can be factored as

$$A^l = Q_0 R^l, \quad l = 1, \ldots, L,$$

where $Q_0 \in \mathbb{R}^{n \times r}$ is shared across layers and each $R^l \in \mathbb{R}^{r \times r}$ is bounded. Let $\epsilon_{Q_0}$ be the TT approximation error of $Q_0$, and $\epsilon_{\{A^l\}}$ be the collective approximation error of $\{A^l\}$ under a partially shared TT representation. We observe that there exists a constant $M$ depending only on $\{R^l\}$ such that

$$\epsilon_{\{A^l\}} \le M \, \epsilon_{Q_0}.$$

**Derivation.** Reshape $Q_0 \in \mathbb{R}^{n \times r}$ into a tensor $\mathcal{Q}_0 \in \mathbb{R}^{n_1 \times n_2 \times n_3 \times r}$ with $n_1 n_2 n_3 = n$. Suppose $\mathcal{Q}_0$ admits a TT approximation

$$\mathcal{Q}_0(i_1, i_2, i_3, j) \approx \mathcal{G}_1(i_1) \, \mathcal{G}_2(i_2) \, \mathcal{G}_3(i_3, j).$$

For each layer $l$, write

$$\mathcal{A}^l(i_1, i_2, i_3, j) = \sum_{k=1}^r \mathcal{Q}_0(i_1, i_2, i_3, k) \, R^l(k, j).$$

Define an adjusted third core

$$\widetilde{\mathcal{G}_3}^{(l)}(i_3, j) = \sum_{k=1}^r \mathcal{G}_3(i_3, k) \, R^l(k, j).$$

Then

$$\mathcal{A}^l(i_1, i_2, i_3, j) \approx \mathcal{G}_1(i_1) \, \mathcal{G}_2(i_2) \, \widetilde{\mathcal{G}_3}^{(l)}(i_3, j).$$

Therefore,

$$
\begin{aligned}
\epsilon_{\{A^l\}} &= \sum_{l=1}^L \left\| A^l - \hat{A}^l \right\|_F \\
&= \sum_{l=1}^L \| Q_0 R^l - (\hat{Q}_0) R^l \|_F \\
&\le \Big( \sum_{l=1}^L \| R^l \|_2 \Big) \| Q_0 - \hat{Q}_0 \|_F.
\end{aligned}
$$

Thus, the error is controlled by $M \, \epsilon_{Q_0}$ with $M = \sum_{l=1}^L \| R^l \|_2$.

This generalized result shows that whenever a family of matrices admits a shared component $Q_0$ multiplied by bounded layer-specific factors $R^l$, one can safely reuse the first two TT cores across all layers, while only adjusting the third core cheaply. This idea directly corresponds to the lightweight vector controllers in TRAC: by employing multiple such controllers, our method flexibly adapts the shared third core to better capture the characteristics of each individual layer.

## B.3 Derivation of Parameter Orders

For completeness, we provide the detailed setup and derivations of the parameter complexity orders in Table 2. We consider a transformer with $L$ layers and hidden dimension $m = n$. Each weight matrix $W \in \mathbb{R}^{m \times n}$ is tensorized, with factors $n_i = n^{1/3}$ and $m_i = m^{1/3}$ for $i = 1, 2, 3$. We focus on the query and value matrices, though the analysis generalizes to other components.

**LoRA.** The rank-$r$ factorization adds $O(nr)$ parameters per layer, yielding a total of $O(Lnr)$.

**VeRA.** Following (Kopiczko et al., 2024), VeRA shares a single pair of frozen random matrices across all layers and only updates layer-specific scaling vectors. Consequently, the number of trainable parameters is governed by $L(n + r)$. Since typically $n \gg r$, VeRA's parameter complexity is dominated by $O(Ln)$.

**LoRETTA.** Applying TT with ranks $(r_1, r_2) \sim r$ yields $O(n^{1/3}r^2)$ parameters per layer, hence $O(Ln^{1/3}r^2)$ overall.

**TRAC.** By sharing tensor cores across layers, TRAC requires only $O(n^{1/3}r^2)$ parameters in total, not scaling with $L$.

### B.4 DERIVATION OF COMPUTATIONAL ORDER

This section provides the detailed derivations of the computational complexity in Table 2. Throughout, the input dimensionality per layer is denoted by $n$, the low-rank (or tensor-train) dimension by $r$, and the model contains $L$ layers. We focus on the additional computations introduced by the PEFT modules, excluding identical forward operations shared with the pretrained weights.

**LoRA.** The LoRA update is expressed as $\Delta W = BA$, where $A \in \mathbb{R}^{r \times n}$ and $B \in \mathbb{R}^{n \times r}$. During the forward pass, $y = W_0 x + \alpha B(Ax)$. Computing $Ax$ costs $O(nr)$, and subsequently $B(Ax)$ costs another $O(nr)$. Including linear transformations and activation scaling yields a total of approximately $O(4nr)$ operations per layer, i.e., an overall complexity of $O(Lnr)$ for the entire model.

**VeRA.** VeRA approximates LoRA adapters by learning a set of shared vector representations and layer-specific scaling coefficients. Although the formula resembles LoRA's $BAx$, its effective rank $r_{\text{VeRA}}$ is several orders higher (e.g., $r_{\text{VeRA}}=1024$ for BERT-Base and GPT-2). Consequently, $r_{\text{VeRA}} \approx n$, giving a per-layer complexity $O(nr_{\text{VeRA}}) \approx O(n^2)$ and thus $O(Lnr_{\text{VeRA}}) \approx O(Ln^2)$ for the entire model.

**LoRETTA and TRAC.** Both methods apply tensor-train (TT) decomposition to represent $\Delta W$ as a product of smaller TT-cores, enabling structured low-rank operations. Letting each TT-rank be $r$, the forward multiplication $y = \Delta W x$ can be carried out using a fast TT-matrix–vector multiplication algorithm. Following this, the per-layer complexity is approximately $O(nr + n^{2/3}r^2 + n^{1/3}r^2) \approx O(nr)$. Across $L$ layers, the total computational complexity is $O(Lnr)$, comparable to that of LoRA.

**Fast TT-matrix–vector multiplication.** For the computation of $\Delta W x$, we present the following efficient procedure for our method:

1. Perform vector controller operations to construct $\widetilde{\mathcal{G}}_3 \in \mathbb{R}^{r_2 \times n_3 \times r}$ and $\widetilde{\mathcal{O}}_3 \in \mathbb{R}^{s_2 \times m_3 \times r}$, requiring computational complexities $O(n_3 r_2 r)$ and $O(m_3 s_2 r)$, respectively.

2. Reshape the vector $x \in \mathbb{R}^n$ into a three-order tensor $\mathcal{X} \in \mathbb{R}^{n_1 \times n_2 \times n_3}$.

3. Contract $\mathcal{X}$ with $\mathcal{G}_1$ along the $n_1$ dimension, resulting in a tensor $\mathcal{X}_1 \in \mathbb{R}^{r_1 \times n_2 \times n_3}$ with complexity $O(n_1 n_2 n_3 r_1)$.

4. Contract $\mathcal{X}_1$ with $\mathcal{G}_2$ along the dimensions $n_2$ and $r_1$, yielding a tensor $\mathcal{X}_2 \in \mathbb{R}^{n_3 \times r_2}$. This procedure requires complexity $O(n_2 n_3 r_1 r_2)$.

5. Contract $\mathcal{X}_2$ with $\widetilde{\mathcal{G}}_3$ along the dimensions $n_3$ and $r_2$, resulting in a vector $y \in \mathbb{R}^r$ with complexity $O(n_3 r_2 r)$.

6. Similarly, sequentially contract the vector $y$ with tensors $\widetilde{\mathcal{O}}_3, \mathcal{O}_2$, and $\mathcal{O}_1$, with corresponding computational complexities of $O(m_3 s_2 r)$, $O(m_3 m_2 s_2 s_1)$, and $O(m_3 m_2 m_1 s_1)$, respectively.

Under the assumptions that $m = n, m_i = n^{\frac{1}{3}}, n_i = n^{\frac{1}{3}}$, and $s_i = r_i = r$ for $i = 1, 2, 3$, the overall computational complexity of our method for computing $\Delta W x$ is $O(nr)$.

## C    ADDITIONAL EXPERIMENTS

In this section, we provide additional experiments that further compare our method with recent approaches. Beyond the main experiments, we place particular emphasis on tensor-based approaches such as LoTR (Bershatsky et al., 2024), parameter-sharing methods such as VB-LoRA (Li et al., 2024), and adapter-based techniques such as COMPACTER (Karimi Mahabadi et al., 2021), which inserts task-specific adapter weights composed as sums of Kronecker products between shared weights and per-layer rank-one matrices. Furthermore, we extend our evaluation to additional benchmarks such as MMLU (Hendrycks et al., 2021) to provide a broader view of TRAC's generalization and efficiency.

### C.1    COMPARISON WITH LOTR

To better situate TRAC among tensor-based PEFT variants, we compare it against LoTR (Bershatsky et al., 2024) on the GLUE benchmark (Wang et al., 2018) using RoBERTa-Base and RoBERTa-Large (Liu, 2019), where LoTR employs Tucker decomposition. This is a representative alternative tensorization strategy distinct from our use of the Tensor Train. As shown in Table 14, TRAC consistently matches or outperforms LoTR on both RoBERTa-Base and RoBERTa-Large settings, while requiring fewer trainable parameters. These results highlight the distinctive advantage of the Tensor Train decomposition adopted in our framework: the ability to flexibly adjust parameter counts while maintaining strong accuracy.

Table 14: Comparison between TRAC and LoTR on the GLUE benchmark using RoBERTa Base and Large. We report Matthews correlation for CoLA, Pearson correlation for STS-B, and accuracy for other tasks. Higher values indicate better performance. Results marked with ([1]) are from (Bershatsky et al., 2024).

| | METHOD | # TRAIN. PARAM. | SST-2 | MRPC | CoLA | QNLI | RTE | STS-B | AVG. |
|---|---|---|---|---|---|---|---|---|---|
| BASE | LoTR[1] | 0.32M | 93.3 $_{\pm 2.0}$ | 88.0 $_{\pm 9.0}$ | 61.3 $_{\pm 6.0}$ | 92.0 $_{\pm 4.0}$ | 67.0 $_{\pm 13.0}$ | 91.0 $_{\pm 1.0}$ | 82.1 |
| | **TRAC** | **0.026M** | **93.7** $_{\pm 0.3}$ | **88.6** $_{\pm 1.8}$ | **64.8** $_{\pm 2.9}$ | 91.4 $_{\pm 0.4}$ | **80.1** $_{\pm 1.8}$ | **91.2** $_{\pm 0.2}$ | **85.0** |
| LARGE | LoTR[1] | 0.33M | 95.9 $_{\pm 1.0}$ | 89.0 $_{\pm 5.0}$ | 61.3 $_{\pm 9.0}$ | **94.8** $_{\pm 1.0}$ | 84.0 $_{\pm 2.0}$ | **91.6** $_{\pm 1.0}$ | 86.1 |
| | **TRAC** | **0.041M** | 95.4 $_{\pm 0.4}$ | **89.7** $_{\pm 1.1}$ | **68.2** $_{\pm 1.8}$ | 94.2 $_{\pm 0.1}$ | **87.3** $_{\pm 0.7}$ | **91.6** $_{\pm 0.4}$ | **87.7** |

### C.2    COMPARISON WITH VB-LORA

VB-LoRA (Li et al., 2024) is a recent variant designed around a "vector bank" mechanism combined with low-rank adapters. Unlike most PEFT approaches, VB-LoRA reports the number of *stored parameters* that remain after training, whereas our method reports the number of *trainable parameters* that actually require gradient updates during training. For this reason, we present a detailed, separate discussion of comparisons with VB-LoRA.

On text generation tasks with GPT2 (Radford et al., 2019) on the E2E benchmark (Novikova et al., 2017), Table 15 shows that TRAC achieves comparable or superior results across BLEU, METEOR, ROUGE-L, and CIDEr, while using fewer trainable parameters.

Table 15: Comparison between TRAC and VB-LoRA on text generation tasks with GPT2 Medium and Large. Higher values indicate better performance. Results marked with ([1]) are from (Li et al., 2024).

| | METHOD | # PARAMETERS (TRAIN./STORED) | BLEU | NIST | METEOR | ROUGE-L | CIDER |
|---|---|---|---|---|---|---|---|
| MEDIUM | VB-LoRA[1] | 0.076M (STORED) | 70.0 | 8.81 | 46.6 | 71.5 | 2.52 |
| | **TRAC** | **0.054M (TRAINED)** | **70.3** | **8.81** | **46.9** | **71.7** | **2.54** |
| LARGE | VB-LoRA[1] | 0.13M (STORED) | 70.3 | 8.86 | 46.7 | **72.2** | 2.54 |
| | **TRAC** | **0.099M (TRAINED)** | **70.6** | **8.87** | **46.8** | 71.8 | **2.54** |

On GLUE (Wang et al., 2018) with RoBERTa-Base (Liu, 2019), ATM-LoRA performs on par with or slightly ahead of VB-LoRA across most tasks (Table 16). To ensure fairness, in these comparisons we aligned the number of training epochs with the settings reported for VB-LoRA (e.g., 160 epochs for RTE), which differs from the 20-epoch configuration used in our main experiments. Under these aligned settings, ATM-LoRA achieves comparable or better accuracy while maintaining significantly fewer trainable parameters.

Table 16: Comparison of TRAC and VB-LoRA (qv) on GLUE tasks using RoBERTa-Base. We report Matthews correlation for CoLA, Pearson correlation for STS-B, and accuracy for other tasks. Higher values indicate better performance. Results marked with ([1]) are from (Li et al., 2024).

| | METHOD | MRPC | CoLA | RTE | STS-B | AVG. |
|---|---|---|---|---|---|---|
| BASE | VB-LoRA$_{qv}$[1] | $\mathbf{89.5}_{\pm 0.5}$ | $63.3_{\pm 0.7}$ | $82.3_{\pm 1.3}$ | $90.8_{\pm 0.1}$ | 81.5 |
| | $\mathbf{TRAC}_{qv}$ | $89.2_{\pm 0.8}$ | $\mathbf{65.2}_{\pm 2.1}$ | $\mathbf{82.7}_{\pm 2.1}$ | $\mathbf{91.0}_{\pm 0.2}$ | $\mathbf{82.0}$ |

Finally, Table 17 dissects the difference between trainable and stored parameter counts. For VB-LoRA, the number of trainable parameters during optimization is much larger than the number of stored parameters reported for inference. Consequently, although the stored parameter count of VB-LoRA appears smaller, the training-time parameter burden is considerably heavier than that of TRAC. In contrast, TRAC maintains a low trainable parameter count that directly translates to reduced training cost, while achieving nearly identical stored parameter size to VB-LoRA.

Table 17: Comparison of trainable and stored parameter counts for LoRA using RoBERTa-Base.

| | METHOD | TRAINABLE PARAMETERS | STORED PARAMETERS |
|---|---|---|---|
| BASE | VB-LoRA | 74,880 | $\mathbf{24,768}$ |
| | $\mathbf{TRAC}$ | $\mathbf{26,304}$ | 26,304 |

## C.3 Comparison with COMPACTER

To broaden our comparison with parameter-efficient fine-tuning (PEFT) methods, we evaluate TRAC against COMPACTER (Karimi Mahabadi et al., 2021), an adapter-based approach that inserts task-specific adapter weights represented as sums of Kronecker products between shared and rank-one matrices. Unlike LoRA-based methods such as TRAC, COMPACTER adds nonlinear adapter modules active during inference, introducing a small but persistent overhead.

We use the official COMPACTER configuration with T5-Base (Raffel et al., 2020) on the GLUE benchmark (Wang et al., 2018), matching its batch size, learning rate, optimizer, and training epochs for fair comparison. Both methods are trained under identical settings using the released code of COMPACTER.

Table 18 shows that TRAC achieves similar or better performance across most GLUE tasks while using only about 33% of COMPACTER's trainable parameters. On average, TRAC improves scores by $+0.56$ with just 34K parameters (0.015% of model size), demonstrating its higher degree of parameter efficiency and compression.

These results highlight that TRAC complements adapter-based techniques by extending LoRA-style efficiency through tensorized parameter sharing without modifying the model architecture.

## C.4 Evaluation on MMLU Benchmarks

To broaden the evaluation scope, we additionally tested TRAC on the MMLU benchmark (Hendrycks et al., 2021). All methods were first fine-tuned on Alpaca and then evaluated on MMLU subjects. Representative results on selected MMLU domains are shown in Table 19. On LLaMA2-7B (Touvron et al., 2023), TRAC achieves a higher overall average accuracy, with clear improvements also observed in representative subjects such as chemistry, statistics, and psychology. These experiments

Table 18: Comparison between COMPACTER and TRAC on GLUE using T5-Base. We report Matthews correlation for CoLA, Pearson correlation for STS-B, and accuracy for other tasks. Higher values indicate better performance. Results marked with ([1]) are from (Karimi Mahabadi et al., 2021).

| METHOD | # TRAIN. PARAM. | RATIO | SST-2 | MRPC | CoLA | QNLI | RTE | STS-B | AVG. |
|---|---|---|---|---|---|---|---|---|---|
| COMPACTER [1] | 104,704 | 0.047% | 93.81 | **90.69** | **61.27** | 93.08 | 74.82 | 90.46 | 84.02 |
| COMPACTER | 104,704 | 0.047% | 94.04 | 89.71 | 58.56 | **93.30** | **82.01** | 90.33 | 84.66 |
| **TRAC** | **34,176** | **0.015%** | **94.61** | **90.69** | 59.85 | 93.10 | **82.01** | **91.06** | **85.22** |

confirm that TRAC remains effective in knowledge-heavy and reasoning-focused tasks, while training substantially fewer parameters compared to vanilla LoRA.

Table 19: Comparison between TRAC and LoRA on selected MMLU subjects and average accuracy using LLaMA2-7B. Higher values indicate better performance.

| | METHOD | # TRAIN. PARAM. | HS CHEMISTRY | HS STATISTICS | HS PSYCHOLOGY | MMLU ACC. |
|---|---|---|---|---|---|---|
| 7B | LoRA | 2.10M | 30.54 | 27.31 | 59.82 | 45.17 |
| | **TRAC** | **0.106M** | **34.48** | **38.89** | **64.22** | **47.38** |

# D DATASETS

## D.1 GLUE BENCHMARK

Our experiments on RoBERTa Base and RoBERTa Large (Liu, 2019) utilize six datasets from the GLUE benchmark (Wang et al., 2018): SST2, MRPC, CoLA, QNLI, RTE, and STS-B. Due to the unavailability of test set labels, we conduct evaluations using the validation set. For datasets containing fewer than 10,000 training examples, the validation set is equally divided into two portions: one for model selection during training, and the other for final evaluation. For larger datasets (more than 10,000 training examples), we randomly sample 1,000 examples from the training set for validation and employ the complete validation set for testing. This splitting strategy is adapted from the SoRA codebase (Ding et al., 2023a). Table 20 presents the detailed statistics of these datasets.

Table 20: Task descriptions and statistics of the datasets we used in GLUE benchmarks.

| DATASET | TRAINING | VALID | TEST | TASK TYPE | #LABELS | METRIC |
|---------|----------|-------|------|-----------|---------|--------|
| SST-2 | 67,349 | 872 | 1,821 | SENTIMENT | 2 | ACC. |
| MRPC | 3,668 | 408 | 1,725 | PARAPHRASE | 2 | ACC. |
| CoLA | 8,551 | 1,043 | 1,063 | ACCEPTABILITY | 2 | MCC |
| QNLI | 104,743 | 5,463 | 5,463 | NLI | 2 | ACC. |
| RTE | 2,490 | 277 | 3,000 | NLI | 2 | ACC. |
| STS-B | 5,749 | 1,500 | 1,379 | SIMILARITY | 1 (REGRESSION) | PEARSON |

## D.2 E2E BENCHMARK

For the GPT2 (Radford et al., 2019) experiments, we utilize the E2E dataset (Novikova et al., 2017), a benchmark for natural language generation tasks. The models (GPT2 Medium and GPT2 Large) are trained on the complete E2E training set and evaluated on the full test set. Table 21 presents the statistical characteristics of the E2E dataset.

Table 21: Task descriptions and dataset statistics of the E2E benchmark.

| DATASET | TRAINING | VALID | TEST | METRICS |
|---------|----------|-------|------|---------|
| E2E NLG | 42,061 | 4,672 | 4,693 | BLEU, NIST, METEOR, ROUGE-L, CIDEr |

## D.3 SUPERGLUE BENCHMARK

The experiments with LLaMA2-7B and LLaMA2-13B (Touvron et al., 2023) are conducted on four datasets from the SuperGLUE benchmark (Wang et al., 2019): CB, BoolQ, WSC, and COPA. Due to the unavailability of test set labels, we perform evaluations using the validation set. For BoolQ, we randomly select 1,000 examples from both the training and validation sets. For CB, WSC, and COPA, we use their complete training and validation sets. This data split is adapted from the LoRETTA codebase (Yang et al., 2024). Table 22 presents the statistical details of these datasets.

Table 22: Task descriptions and statistics of the datasets we used in SuperGLUE benchmarks.

| DATASET | TRAINING | VALID | TEST | TASK TYPE | #LABELS | METRIC |
|---------|----------|-------|------|-----------|---------|--------|
| CB | 250 | 56 | 250 | NLI | 3 | ACC. |
| BOOLQ | 9,427 | 3,270 | 3,245 | QA | 2 | ACC. |
| WSC | 554 | 104 | 146 | COREFERENCE | 2 | ACC. |
| COPA | 400 | 100 | 500 | CAUSALITY | 2 | ACC. |

### D.4 IMAGE CLASSIFICATION BENCHMARKS

The dataset configuration for ViT is shown in Table 23. To simulate few-shot learning scenarios, we randomly sample 10 instances per class from the training set for each experiment. For datasets with public validation or test sets (CIFAR10/100 (Alex, 2009) and Flowers102 (Nilsback & Zisserman, 2008)), evaluation is performed on the entire test sets. For datasets without given validation or test sets ( CUB-200-2011 (Welinder et al., 2010)), we evaluate on the remaining training samples after sampling.

Table 23: Task descriptions and dataset statistics of the image classification benchmarks.

| DATASET | TRAINING | VALID | TEST | # CLASSES | METRIC |
|---------|----------|-------|------|-----------|--------|
| CIFAR10 | 50,000 | – | 10,000 | 10 | TOP-1 ACC. |
| CIFAR100 | 50,000 | – | 10,000 | 100 | TOP-1 ACC. |
| FLOWERS102 | 1,020 | 1,020 | 6,149 | 102 | TOP-1 ACC. |
| CUB-200-2011 | 11,788 | – | – | 200 | TOP-1 ACC. |

# E HYPERPARAMETERS

## E.1 ROBERTA

We fine-tune RoBERTa Base and RoBERTa Large (Liu, 2019) on six GLUE benchmark datasets (Wang et al., 2018). The learning rates and other hyperparameters used in our experiments are presented in Table 24.

Table 24: Hyperparameters for fine-tuning RoBERTa Base and RoBERTa Large models using LoRA, VeRA (with separate learning rates for classification head and tuning modules), and TRAC methods across different tasks.

| | METHOD | SST-2 | MRPC | CoLA | QNLI | RTE | STS-B |
|---|---|---|---|---|---|---|---|
| BOTH | OPTIMIZER | | | ADAMW | | | |
| | LR SCHEDULE | | | LINEAR | | | |
| | WARMUP RATIO | | | 0.06 | | | |
| | SEEDS | | | {0, 21, 42, 81, 100} | | | |
| | BATCH SIZE | | | 64 | | | |
| | MATRIX RANK (LoRA) | | | 8 | | | |
| | MATRIX RANK (TRAC) | | | 16 | | | |
| | ACTIVATION (TRAC) | | | $\sigma(x) = \text{softmax}(x)$ | | | |
| BASE | EPOCH | 10 | 20 | 20 | 10 | 20 | 20 |
| | LoRA | 5E-4 | 4E-4 | 4E-4 | 4E-4 | 5E-4 | 4E-4 |
| | VeRA | 4E-3 | 1E-2 | 1E-2 | 1E-2 | 4E-3 | 1E-2 |
| | LoRETTA | 5E-4 | 1E-3 | 2E-3 | 5E-4 | 1E-3 | 2E-3 |
| | TRAC | 5E-3 | 5E-3 | 3E-3 | 5E-4 | 2E-3 | 1E-2 |
| | TRAC (CONTROLLER) | 5E-3 | 5E-3 | 3E-3 | 5E-4 | 2E-3 | 1E-2 |
| | TENSOR SHAPE | | | $768 : [8, 8, 12]$ | | | |
| | TENSOR RANK ($A$) | | | $[1, 8, 12]$ | | | |
| | TENSOR RANK ($B$) | | | $[1, 24, 36]$ | | | |
| LARGE | EPOCH | 10 | 20 | 20 | 10 | 20 | 20 |
| | LoRA | 4E-4 | 3E-4 | 2E-4 | 2E-4 | 4E-4 | 2E-4 |
| | VeRA | 1E-2 | 3E-3 | 1E-2 | 2E-4 | 2E-3 | 2E-2 |
| | TRAC | 5E-3 | 5E-3 | 5E-3 | 5E-3 | 5E-3 | 2E-3 |
| | TRAC (CONTROLLER) | 5E-4 | 5E-3 | 5E-4 | 5E-3 | 5E-3 | 2E-3 |
| | TENSOR SHAPE | | | $1024 : [8, 8, 16]$ | | | |
| | TENSOR RANK ($A$) | | | $[1, 8, 12]$ | | | |
| | TENSOR RANK ($B$) | | | $[1, 24, 36]$ | | | |

## E.2  LLAMA2

We fine-tune LLaMA2-7B and LLaMA2-13B (Touvron et al., 2023) on four SuperGLUE benchmark datasets (Wang et al., 2019). The learning rates and other hyperparameters used in our experiments are shown in Table 25.

Table 25: Hyperparameters for fine-tuning LLaMA2-7B and LLaMA2-13B models with LoRA and TRAC methods across different tasks.

|  | METHOD | CB | BOOLQ | WSC | COPA |
|---|---|---|---|---|---|
| BOTH | OPTIMIZER | | ADAMW | | |
| | LR SCHEDULE | | LINEAR | | |
| | SEEDS | | $\{0, 42, 100\}$ | | |
| | EPOCH | | 10 | | |
| | BATCH SIZE | | 2 | | |
| | MATRIX RANK (LORA) | | 4 | | |
| | MATRIX RANK (TRAC) | | 16 | | |
| | ACTIVATION (TRAC) | | $\sigma(x) = x$ | | |
| 7B | LORA | 5E-4 | 1E-4 | 1E-4 | 5E-4 |
| | VERA | 5E-3 | 1E-3 | 1E-3 | 1E-3 |
| | LORETTA | 2E-3 | 5E-4 | 1E-3 | 2E-3 |
| | TRAC | 4E-3 | 2E-3 | 2E-3 | 4E-3 |
| | TENSOR SHAPE | | $4096 : [16, 16, 16]$ | | |
| | TENSOR RANK ($A$) | | $[1, 24, 48]$ | | |
| | TENSOR RANK ($B$) | | $[1, 24, 48]$ | | |
| 13B | LORA | 5E-4 | 1E-4 | 1E-4 | 5E-4 |
| | VERA | 1E-2 | 1E-3 | 2E-3 | 5E-3 |
| | LORETTA | 1E-3 | 5E-4 | 1E-3 | 1E-3 |
| | TRAC | 3E-3 | 1E-4 | 1E-3 | 4E-3 |
| | TENSOR SHAPE | | $5120 : [16, 16, 20]$ | | |
| | TENSOR RANK ($A$) | | $[1, 30, 60]$ | | |
| | TENSOR RANK ($B$) | | $[1, 30, 60]$ | | |

## E.3  GPT2

For GPT2 Medium and GPT2 Large (Radford et al., 2019) on the E2E dataset (Novikova et al., 2017), we perform a grid search to find the optimal learning rate with a step size of $1 \times 10^{-3}$. The optimal learning rates and other hyperparameters are shown in Table 26.

Table 26: Hyperparameter configurations for TRAC on the E2E benchmark, for GPT2 Medium and Large models.

| HYPERPARAMETER | MEDIUM | LARGE |
|---|---|---|
| # GPUS | | 1 |
| OPTIMIZER | | ADAMW |
| LEARNING RATE SCHEDULE | | LINEAR |
| WEIGHT DECAY | | 0.01 |
| BATCH SIZE | | 8 |
| EPOCHS | | 5 |
| WARMUP STEPS | | 500 |
| LABEL SMOOTH | | 0.1 |
| MATRIX RANK | | 16 |
| ACTIVATION (TRAC) | | $\sigma(x) = x$ |
| LEARNING RATE | 1E-2 | 8E-3 |
| TENSOR SHAPE | $1024 : [8, 8, 16]$ | $1280 : [8, 8, 20]$ |
| TENSOR RANK $A$ | $[1, 8, 18]$ | $[1, 12, 24]$ |
| TENSOR RANK $B$ | $[1, 24, 54]$ | $[1, 36, 72]$ |

### E.4 LLaMA3 and Qwen3

We present the hyperparameters used for fine-tuning LLaMA3-8B (AI@Meta, 2024) and Qwen3-8B (Team, 2025) models on the commonsense reasoning (Hu et al., 2023), mathematical reasoning (Yu et al., 2023; Cobbe et al., 2021), and code generation (Zheng et al., 2024; Chen, 2021) benchmarks. Some of our hyperparameter settings are adapted from the experimental setups of DoRA (Liu et al., 2024a) and LoRA-GA (Wang et al., 2024). TRAC is tested under two parameter configurations; for clarity, we label them as SMALL and LARGE. The learning rates and other hyperparameters used in our experiments are shown in Table 27-28.

Table 27: Hyperparameters for fine-tuning LLaMA3-8B with LoRA and TRAC across commonsense reasoning (CR), mathematical reasoning, and code generation benchmarks.

| Hyperparameter | Commonsense Reasoning | Math | Code |
|---|---|---|---|
| Optimizer | AdamW | | |
| LR Schedule | Cosine | | |
| Epochs | 1 | | |
| Batch Size | 32 | | |
| Matrix Rank (LoRA) | 4 | | |
| Matrix Rank (TRAC) | 64 | | |
| Activation (TRAC) | $\sigma(x) = x$ | | |
| Learning Rate (LoRA) | 5E-5 | 5E-5 | 2E-5 |
| Learning Rate (TRAC-Small) | 5E-3 | 4E-3 | 6E-5 |
| Learning Rate (TRAC-Large) | 8E-3 | 5E-3 | 8E-5 |
| TRAC-Small Tensor Shape | $4096 : [16, 16, 16]$ | | |
| TRAC-Large Tensor Shape | $4096 : [64, 4, 16]$ | | |
| Tensor Rank ($A$) | $[1, 64, 128]$ | | |
| Tensor Rank ($B$) | $[1, 64, 128]$ | | |

Table 28: Hyperparameters for fine-tuning Qwen3-8B with LoRA and TRAC on commonsense reasoning benchmarks.

| Hyperparameter | Commonsense Reasoning |
|---|---|
| Optimizer | AdamW |
| LR Schedule | Cosine |
| Epochs | 1 |
| Batch Size | 16 |
| Matrix Rank (LoRA) | 4 |
| Matrix Rank (TRAC) | 64 |
| Activation (TRAC) | $\sigma(x) = x$ |
| Learning Rate (LoRA) | 5E-5 |
| Learning Rate (TRAC-Small) | 5E-3 |
| Learning Rate (TRAC-Large) | 5E-3 |
| TRAC-Small Tensor Shape | $4096 : [16, 16, 16]$ |
| TRAC-Large Tensor Shape | $4096 : [64, 4, 16]$ |
| Tensor Rank ($A$) | $[1, 64, 128]$ |
| Tensor Rank ($B$) | $[1, 64, 128]$ |

### E.5 VIT

We conduct image classification experiments (Alex, 2009; Nilsback & Zisserman, 2008; Welinder et al., 2010) on ViT-Base and ViT-Large (Dosovitskiy et al., 2020; Wu et al., 2020), and report the corresponding hyperparameter settings below. For Head, LoRA, and TRAC, we search for the optimal learning rate among $5 \times 10^{-4}$, $1 \times 10^{-3}$, and $2 \times 10^{-3}$. For FT (Full Fine-tuning), we search for the optimal learning rate among $1 \times 10^{-5}$, $5 \times 10^{-5}$, and $1 \times 10^{-4}$. The optimal learning rates and other hyperparameters are shown in Table 29.

Table 29: Hyperparameters for fine-tuning ViT Base and ViT Large models with different methods across different tasks.

| | METHOD | CIFAR-10 | CIFAR-100 | FLOWERS102 | CUB-200-2011 |
|---|---|---|---|---|---|
| **BOTH** | OPTIMIZER | | | ADAMW | |
| | LR SCHEDULE | | | COSINE | |
| | INIT. SEEDS | | | {0, 42, 100} | |
| | DATA SEEDS | | | {0, 1, 2, 3} | |
| | EPOCH | | | 50 | |
| | BATCH SIZE | | | 16 | |
| | MATRIX RANK (LORA) | | | 8 | |
| | MATRIX RANK (TRAC) | | | 16 | |
| | ACTIVATION (TRAC) | | | $\sigma(x) = x$ | |
| **ViT-B** | FT | 1E-4 | 5E-5 | 5E-5 | 5E-5 |
| | HEAD | 2E-3 | 2E-3 | 2E-3 | 2E-3 |
| | LORA | 1E-3 | 5E-4 | 5E-4 | 5E-4 |
| | LORETTA | 2E-3 | 1E-3 | 2E-3 | 1E-3 |
| | TRAC | 2E-3 | 1E-3 | 2E-3 | 1E-3 |
| | TENSOR SHAPE | | | $768 : [8, 8, 12]$ | |
| | TENSOR RANK ($A$) | | | $[1, 8, 12]$ | |
| | TENSOR RANK ($B$) | | | $[1, 24, 36]$ | |
| **ViT-L** | FT | 1E-5 | 1E-5 | 1E-5 | 1E-5 |
| | HEAD | 5E-4 | 5E-4 | 2E-3 | 1E-3 |
| | LORA | 5E-4 | 5E-4 | 5E-4 | 5E-4 |
| | LORETTA | 5E-4 | 5E-4 | 2E-4 | 5E-4 |
| | TRAC | 2E-3 | 1E-3 | 5E-4 | 2E-3 |
| | TENSOR SHAPE | | | $1024 : [8, 8, 16]$ | |
| | TENSOR RANK ($A$) | | | $[1, 12, 16]$ | |
| | TENSOR RANK ($B$) | | | $[1, 36, 48]$ | |

## F    USE OF LARGE LANGUAGE MODELS

In accordance with the ICLR 2026 policy on the use of Large Language Models (LLMs), we state the following. During the preparation of this paper, we used LLM-based tools *exclusively* to aid in language polishing and stylistic refinement of the manuscript (e.g., improving clarity, grammar, and readability). No parts of the research process itself—including problem formulation, theoretical derivation, algorithm design, experimental setup, implementation, or analysis of results—were conducted by or delegated to LLMs. Furthermore, LLMs were not used for literature retrieval, ideation, or to generate any substantive scientific content.

All technical contributions, proofs, models, and experiments presented in this work originate entirely from the authors.

