# OpenReview forum: "TRAC: Tensor-Train based Across-layer Compression for Parameter-Efficient Fine-Tuning"
_ICLR.cc/2026/Conference — ICLR 2026 Poster_

### Official Review · Reviewer_Sk7c · 2025-10-28

**Soundness:** 3
**Presentation:** 3
**Contribution:** 3
**Rating:** 8
**Confidence:** 5

**Summary:**

This paper introduces TRAC (Tensor-Train based Across-layer Compression), a framework for parameter-efficient fine-tuning (PEFT) of large pre-trained models using Tensor-Train (TT) decomposition. TRAC combines tensor decomposition with across-layer parameter sharing and freezing, incorporating lightweight controllers to enhance adaptability. The paper evaluates TRAC on diverse architectures, including LLaMA-2, GPT, BERT, and ViT, across tasks such as text classification, text generation, and image classification. The results demonstrate that TRAC achieves comparable or superior performance to existing methods (e.g., LoRA, VeRA, and LoRETTA) with significantly fewer trainable parameters. The paper claims a reduction of up to 20× fewer trainable parameters for LLaMA-2-13B on SuperGLUE tasks while maintaining or improving performance.

**Strengths:**

1. The proposed method achieves up to 20× parameter reduction compared to LoRA in certain tasks while maintaining comparable or superior performance. This is particularly impactful for resource-constrained scenarios.
2. TRAC effectively combines Tensor-Train decomposition with cross-layer parameter sharing and freezing, which is a novel and well-motivated design. This approach directly addresses the redundancy in deep transformer layers.
3. The experiments are conducted across various architectures (e.g., BERT, LLaMA, GPT, ViT) and tasks (e.g., GLUE, SuperGLUE, E2E, image classification). The paper includes comparisons with strong baselines like LoRA, VeRA, and LoRETTA, supported by statistical analysis.

**Weaknesses:**

1. While the paper emphasizes the importance of lightweight controllers, the experiments do not deeply explore the trade-offs of different controller designs or activation functions. This limits the understanding of the controller's impact on performance.
2. The paper does not sufficiently discuss some related PEFT  methods, such as AdaLoRA[1], MPOP[2]. Including a broader discussion would enhance the positioning of TRAC in the literature.
3. While the paper claims that higher-order tensors do not improve performance, the exploration of tensor orders beyond third-order is limited to one or two configurations. A more detailed analysis would help justify this design choice.

Ref:
[1]. Zhang Q, Chen M, Bukharin A, et al. Adalora: Adaptive budget allocation for parameter-efficient fine-tuning[J]. arXiv preprint arXiv:2303.10512, 2023.
[2]. Liu P, Gao Z F, Zhao W X, et al. Enabling lightweight fine-tuning for pre-trained language model compression based on matrix product operators[J]. arXiv preprint arXiv:2106.02205, 2021.

**Questions:**

1. Could the authors provide more intuition about why gating shared cores with vectors (vs. low-rank residuals or elementwise modulation) is expressive enough? Have you explored alternative controller architectures (e.g., learned affine transforms, attention-based modulation)?
2. Could the authors include a more explicit guideline or heuristic for selecting tensor shapes under a fixed parameter budget? How sensitive is the performance to suboptimal tensorization choices?
3. Have the authors identified any regimes (e.g., early/late transformer blocks, deeper MoE architectures) where sharing harms performance? Would allowing partial sharing (e.g., blockwise) improve robustness? If not tested, can the authors comment on expected behavior?

---

> ### Author Response · Authors · 2025-11-26
> **Response to Reviewer Sk7c— Part 1/3**
>
> ### R4-Q1. Analysis of controller design and activation functions
>
> *(addressing Weakness 1)*
>
> **A1:** We sincerely thank the reviewer for the insightful comments regarding the **lightweight controller design** and **activation function choices**, which help us clarify our design motivation and their influence within TRAC.
>
> Our motivation originated from the observation that directly sharing tensor cores across layers may lead to performance degradation. To alleviate this issue, we introduced a **layer‑specific lightweight modulation mechanism (controller)** to enhance the flexibility of shared tensor representations across layers.
>
> During design exploration, we evaluated *matrix‑form* and *tensor‑form* controllers; however, both incurred substantial increases in trainable parameters, which contradicted our goal of achieving extreme compression. We therefore adopted a **vector‑form controller**, which preserves essential layer‑wise adaptability while adding negligible parameter overhead.
>
> Inspired by the gating mechanism in Mixture‑of‑Experts (MoE) models, we further employed a **Softmax‑style activation**, enabling the controller to adaptively rescale shared tensor cores based on layer‑specific characteristics.
>
> To more systematically examine the influence of different activation functions, we conducted an ablation study on RoBERTa‑Base, comparing four nonlinearities: ReLU, Tanh, Sigmoid, and Softmax. The results are summarized below.
>
> | TASK    |    MRPC    |    COLA    |    RTE     |   STS‑B    |
> | :------ | :--------: | :--------: | :--------: | :--------: |
> | Softmax | 88.6 (1.8) | 64.8 (2.9) | 80.1 (1.8) | 91.2 (0.2) |
> | ReLU    | 87.8 (0.4) | 64.1 (2.0) | 76.6 (5.5) | 90.6 (0.2) |
> | Tanh    | 87.9 (0.4) | 63.8 (2.0) | 76.3 (5.2) | 90.9 (0.2) |
> | Sigmoid | 87.8 (0.9) | 61.4 (2.6) | 78.0 (3.1) | 90.9 (0.2) |
>
> The results indicate that the **Softmax‑style activation consistently achieves better overall performance**. We attribute this improvement to its exponential normalization, which enhances the dynamic scaling of shared weights and improves model adaptability. The other activation functions yield *slightly lower performance but remain within a comparable range*, without causing drastic degradation.
>
> Going forward, we plan to further investigate different combinations of controller structures and nonlinear activations to deepen our understanding of how modulation mechanisms influence shared‑parameter performance.
>
> ### R4-Q2. Discussion of missing related PEFT methods (AdaLoRA, MPOP)
>
> *(addressing Weakness 2)*
>
> **A2:** We sincerely thank the reviewer for the constructive suggestion to expand the discussion of related PEFT methods. Below we briefly summarize two representative approaches and clarify their relation to TRAC.
>
> - **AdaLoRA** dynamically adjusts the LoRA rank during training through an adaptive capacity allocation mechanism across layers, enabling more flexible parameter usage. While AdaLoRA focuses on **adaptive rank optimization**, **TRAC** targets **extreme parameter reduction** via tensorization and cross‑layer sharing—two complementary directions for improving LoRA‑style methods.
>
> - **MPOP** represents model weights using the Matrix‑Product‑Operator (MPO) (equivalent to the Tensor‑Train decomposition) to **compress pretrained models**, which are then fine‑tuned. In contrast, **TRAC** directly performs parameter‑efficient fine‑tuning on the pretrained backbone by adding tensor modules, addressing a different adaptation scenario.
>
> Both methods provide valuable insights into the PEFT and tensorization literature. We will include explicit citations and an extended discussion of AdaLoRA and MPOP in the revised version to better clarify TRAC’s technical positioning.

---

> ### Author Response · Authors · 2025-11-26
> **Response to Reviewer Sk7c— Part 2/3**
>
> ### R4-Q3. Analysis of higher‑order tensors and tensor‑shape selection
>
> *(addressing Weakness 3 and Question 2)*
>
> **A3:** We thank the reviewer for raising this insightful question about tensor order and shape selection. Indeed, decomposing the LoRA matrices into Tensor‑Train (TT) form introduces extra degrees of freedom—namely the tensor *order*, *shape*, and *TT‑rank*.In our experiments, we explored these factors under a roughly **constant total parameter budget**, and we summarize our main observations and practical guidelines below.
>
> ### 1. Impact of Higher‑Order Tensors
>
> To systematically examine higher‑order configurations, we evaluated 3‑, 4‑, and 5‑order tensors under a constant parameter budget and compared their performance **across different freezing/sharing strategies**.
>
> **Table 3-A. Standard configuration for 3‑order tensor**
>
> | Configuration               | # Params | COLA (%)   | MRPC (%)   | RTE (%)    | STS‑B (%)  |
> | --------------------------- | -------- | ---------- | ---------- | ---------- | ---------- |
> | [Trainable, Frozen, Shared] | 26,304   | 64.8 (2.9) | 88.6 (1.8) | 80.1 (1.8) | 91.2 (0.2) |
>
> **Table 3-B. Additional configurations for 4‑order tensor**
>
> | Configuration                       | # Params | COLA (%)   | MRPC (%)   | RTE (%)    | STS‑B (%)  |
> | ----------------------------------- | -------- | ---------- | ---------- | ---------- | ---------- |
> | [Trainable, Frozen, Frozen, Shared] | 26,784   | 64.5 (1.2) | 87.5 (1.4) | 75.3 (2.8) | 90.8 (0.2) |
> | [Trainable, Shared, Frozen, Shared] | 28,480   | 64.0 (1.8) | 87.6 (1.4) | 78.7 (3.4) | 90.7 (0.3) |
> | [Trainable, Frozen, Shared, Shared] | 28,416   | 62.9 (1.9) | 86.5 (0.5) | 76.1 (2.3) | 90.7 (0.3) |
>
> **Table 3-C. Additional configurations for 5‑order tensor**
>
> | Configuration                               | # Params | COLA (%)   | MRPC (%)   | RTE (%)    | STS‑B (%)  |
> | ------------------------------------------- | -------- | ---------- | ---------- | ---------- | ---------- |
> | [Trainable, Frozen, Frozen, Frozen, Shared] | 28,560   | 63.7 (1.5) | 88.6 (0.6) | 76.4 (3.1) | 90.7 (0.2) |
> | [Trainable, Shared, Frozen, Frozen, Shared] | 28,376   | 65.8 (1.0) | 88.4 (0.9) | 78.6 (1.6) | 90.9 (0.2) |
> | [Trainable, Frozen, Frozen, Shared, Shared] | 28,464   | 62.9 (2.3) | 88.5 (1.0) | 77.4 (2.6) | 90.4 (0.3) |
> | [Trainable, Frozen, Shared, Frozen, Shared] | 28,592   | 62.6 (2.1) | 89.0 (1.4) | 78.4 (1.2) | 91.1 (0.1) |
>
> We observe that higher‑order configurations lead to **moderate variations or decreases** in performance compared to the standard 3‑order setup. Nevertheless, the results *remain within a comparable range*, indicating that TRAC *maintains reasonably consistent performance* even under sub‑optimal tensor configurations.
>
> From a practical perspective, considering performance, efficiency, and simplicity, we therefore recommend the **3‑order configuration** as a balanced setup.
>
> ### 2. Empirical Guidelines for Tensor Shape and TT‑Rank Selection
>
> To achieve stable performance under a fixed parameter budget, we summarize the following empirical heuristics (taking the 3‑order case as an example):
>
> - **Tensor Shape Selection:**
>   For a given hidden dimension $n$, choose dimensions that are approximately equal, satisfying $n_1 \approx n_2 \approx n_3 \approx n^{1/3}$.
>   The last dimension is typically set slightly larger to align with shared modules. Example configurations include 768 → [8, 8, 12], 1024 → [8, 8, 16], and 4096 → [16, 16, 16].
>
> - **Rank Assignment:**
>   Start with symmetric ranks $r_1 = r_2 = r_3 = r$, estimated from the approximate parameter formula
>   $$
>   P \approx L(n_1r_1 + r_2 + n_3) + n_3r_2r,
>   $$
>   and then adjust $r_1$ and $r_3$ to control capacity distribution. Increasing $r_1$ improves layer‑specific flexibility, while increasing $r_3$ enhances the cross‑layer reusability of shared cores.
>
> We will include in the revised appendix **recommended tensor configurations** for different model scales.

---

> ### Author Response · Authors · 2025-11-26
> **Response to Reviewer Sk7c— Part 3/3**
>
> ### R4-Q4. Expressivity and alternatives of the vector‑based controller
>
> *(addressing Question 1)*
>
> **A4:** We sincerely thank the reviewer for raising this insightful question. Our design objective was to maintain **layer‑specific adaptability under an extremely tight parameter budget**. Based on this goal, we adopted a **vector‑based gating mechanism**, motivated by the following considerations:
>
> 1. **Structural simplicity and parameter efficiency:**
>    The vector‑based gating controller introduces only two small vectors per layer, each with dimensions typically ≤ 32, resulting in a significantly lower parameter count than matrix‑ or tensor‑form gating. This design adds negligible training and storage overhead, preserving the overall lightweight nature of TRAC.
>
> 2. **Sufficient expressive capability:**
>    Despite its minimal structure, the vector‑based gating enables fine‑grained modulation of shared tensor cores across layers. Our ablation studies show that incorporating this gating effectively mitigates the performance degradation observed when cores are shared directly.
>
> We greatly appreciate the reviewer’s suggestions regarding alternative controller architectures such as *low‑rank residuals*, *element‑wise modulation*, *learned affine transforms*, and *attention‑based modulation*. These design directions are highly inspiring and have strong potential to further enhance the adaptability of shared structures. We plan to further explore these options in future work to better understand the performance‑efficiency trade‑offs of different control mechanisms within PEFT scenario.
>
> ### R4-Q5. Further exploration of parameter sharing mechanisms
>
> *(addressing Question 3)*
>
> **A5:**  We sincerely thank the reviewer for this insightful question on the applicability and potential limitations of the sharing mechanism. Our current observations and hypotheses are summarized as follows:
>
> 1. **Early vs. Late Transformer Blocks:**
>    Deeper (late) blocks typically capture higher‑level semantic abstractions. Excessive sharing in these layers may constrain their representational flexibility and lead to slight performance degradation. In contrast, earlier blocks primarily encode more general features, where sharing exerts a much smaller influence on performance.
>
> 2. **Potential in MoE Architectures:**
>    We plan to extend the sharing mechanism to parameters across corresponding layers of multi‑expert (MoE) models. Intuitively, experts situated at the same layer position are likely to exhibit redundancy. Sharing part of their parameters could further compress the overall model size and possibly reduce unnecessary duplication among experts.
>
> 3. **Partial (blockwise) Sharing and Robustness:**
>    Although we have not yet evaluated this setting, our current evidence suggests that blockwise sharing may strengthen parameter correlations, which could in turn reduce robustness to input perturbations or distribution shifts. Introducing moderate random freezing or adversarial regularization could potentially alleviate this effect while retaining high compression ratios.
>
> Looking forward, we plan to systematically study the sharing strategies in complex scenarios and heterogeneous architectures. This will allow us to better characterize the trade‑off between parameter sharing and model performance, and to derive practical design guidelines for future PEFT approaches.

---

> ### Comment · Reviewer_Sk7c · 2025-11-27
> **More Question.**
>
> After reading the comments from other reviewers, I have a new concern. Please clarify the differences and innovations of the proposed method compared to COMPACTER.

---

> ### Author Response · Authors · 2025-11-29
> **Response to the reviewer’s concern about the comparison between TRAC and COMPACTER**
>
> ### R4‑Q6. Differences and innovations compared to COMPACTER
>
> **A6:** We sincerely thank the reviewer for raising this valuable and insightful question. We fully recognize the importance of **COMPACTER** [1] in advancing the field of parameter‑efficient fine‑tuning (PEFT). While both TRAC and COMPACTER explore **parameter sharing** to reduce the number of trainable parameters, TRAC fundamentally differs from COMPACTER in **research direction**, **technical foundation**, and **modeling mechanisms**, as detailed below.
>
> ### (1) Distinct methodological paradigms and modeling lines
>
> The PEFT literature comprises several parallel paradigms, such as *Prompt‑based*, *Adapter‑based*, and *LoRA‑based* approaches.
> **COMPACTER** follows the *adapter‑based* paradigm, inserting nonlinear bottleneck modules into selected layers to achieve parameter‑efficient fine‑tuning. Such modules must remain active during inference, altering the pretrained architecture and introducing inference‑time overhead.
>
> In contrast, **TRAC** builds upon the *LoRA‑based* paradigm, directly applying low‑rank perturbations in the weight space. The adapted weights can be fully merged with the pretrained ones before inference, preserving the model architecture and avoiding any additional inference latency.
>
> These two approaches therefore belong to parallel technical families. **TRAC** represents an extension of parameter‑sharing and freezing mechanisms **within the LoRA paradigm**, serving as an independent yet complementary line of research relative to **COMPACTER**.
>
> ### (2) Different mathematical tools for parameter compression
>
> **COMPACTER** employs *Kronecker product* factorization to construct Low‑rank Parameterized Hypercomplex Multiplication (LPHM) layers and compresses the parameters by sharing portions of the Kronecker factors.
>
> In contrast, **TRAC** adopts *Tensor‑Train (TT) decomposition* and shares selected TT cores across similar types of weight matrices within Transformer architectures.
>
> These two decompositions differ substantially in their mathematical properties and expressive characteristics. **TRAC** systematically explores efficient modeling and optimization strategies for shared‑core TT decomposition in PEFT settings—an approach orthogonal to **COMPACTER**’s Kronecker‑based modeling, and thus constitutes a **technically independent innovation**.
>
> ### (3) Additional mechanisms and innovations in TRAC
>
> Beyond the decomposition and sharing scheme itself, **TRAC** further introduces the following design elements—**which extend beyond the scope of COMPACTER**—to enhance parameter efficiency and flexibility:
>
> - **Randomized freezing mechanism:** randomly initializes and freezes a subset of TT cores during fine‑tuning;
> - **Lightweight layer‑wise controller:** dynamically adjusts layer‑specific variations on top of shared parameters, balancing generalization and expressivity;
> - **Asymmetric parameter allocation:** exploits the intrinsic functional asymmetry between LoRA’s $A$ and $B$ matrices to achieve better performance under the same parameter budget.
>
> Empirically, TRAC achieves a **notably higher compression ratio** while maintaining model performance—training only 0.021 % of total parameters on RoBERTa‑Base and 0.0016 % on LLaMA‑2‑7B, compared to 0.047 % for COMPACTER.
>
> ### (4) Empirical comparison with COMPACTER
>
> To further validate these distinctions, we performed a direct comparison between **TRAC** and **COMPACTER** using the *official COMPACTER codebase* under the same experimental setup as [1], with T5‑Base evaluated on GLUE tasks.
>
> ***Table  6. Comparison between  COMPACTER  and  TRAC***
> |Method|# Param.|Ratio|SST‑2|MRPC|CoLA|QNLI|RTE|STS‑B|Avg.|
> |-|-:|-:|-:|-:|-:|-:|-:|-:|-:|
> |COMPACTER [1]|104,704|0.047 %|93.81|**90.69**|**61.27**|93.08|74.82|90.46|84.02|
> |COMPACTER (Reproduced)|104,704|0.047 %|94.04|89.71|58.56|**93.30**|**82.01**|90.33|84.66|
> |**TRAC**|**34,176**|**0.015 %**|**94.61**|**90.69**|59.85|93.10|**82.01**|**91.06**|**85.22**|
>
> As shown above, **TRAC** achieves comparable or better performance with only **about 33 % of COMPACTER’s trainable parameters**, indicating a higher degree of parameter efficiency and compression.
>
> ### Summary
>
> In summary, while both **TRAC** and **COMPACTER** utilize parameter sharing for model compression, they **differ substantially** in their **methodological paradigms** (*LoRA‑based vs. Adapter‑based*), **mathematical formulations** (*Tensor Train vs. Kronecker product*), and **underlying design targets**, highlighting that **TRAC** constitutes an **independent and complementary research line** in PEFT.
>
> We sincerely thank the reviewer for this insightful comment, which has helped us further clarify the distinctions and empirical positioning of our approach relative to **COMPACTER**. In the revised version, we will make these differences more explicit to improve clarity.
>
> [1] Karimi Mahabadi et al., *“COMPACTER: Efficient Low‑Rank Hypercomplex Adapter Layers,”* NeurIPS 2021.

---

### Official Review · Reviewer_Afac · 2025-10-30

**Soundness:** 3
**Presentation:** 4
**Contribution:** 2
**Rating:** 6
**Confidence:** 4

**Summary:**

The paper studies a new PEFT adapter based on tensor-train format initialization. Compared with widely used PEFT methods like LoRA, the proposed method greatly reduces the number of trainable parameters. Compared with existing tensor-train–based adapters, the proposed method further includes freezing and sharing designs to achieve better performance without compromising parameter efficiency. Experimental results on the BERT, LLaMA, and ViT model families demonstrate the effectiveness of the proposed method in balancing efficiency and performance.

**Strengths:**

- The paper is well-written and provides sufficient background knowledge on the tensor-train format, making it easy to follow.
- In addition to NLP tasks, the paper also considers vision classification tasks to demonstrate the effectiveness of the proposed method.
- The paper provides theoretical analysis on the effectiveness of the TRAC method.

**Weaknesses:**

- Even though the proposed method has not been explored before, there is sufficient prior work on freezing (e.g., AFLoRA) or sharing (e.g., ShareLoRA) parts of adapters. This work appears to mainly combine these ideas, which limits its overall contribution.
- I wonder if the authors could provide a more direct explanation for the motivation behind studying an even more parameter-efficient method than LoRA. In my opinion, further reducing parameters may not significantly decrease the training cost compared with standard LoRA methods.
- I am satisfied with the experiments on the BERT and ViT models. However, for the experiments on the LLaMA-2 model, the tested tasks seem too limited. I wonder if the authors have evaluated the proposed method on more challenging reasoning tasks, such as those used in the DoRA paper. These tasks might better align with today’s real-world PEFT setups. Additionally, all models considered in this paper are somewhat outdated; I would expect some experiments on more recent models.
- I am quite interested in the results regarding the symmetry analysis in Appendix A.3. Could the authors provide more intuition behind these results? Are there similar findings in LoRA-style works?

**Questions:**

- In the experiments, the authors set the LoRA rank to 4. However, I recall that the optimal setup for LoRA rank is typically 8 or 16. Would the performance gap between LoRA and TRAC increase if a higher LoRA rank were used in Tables 3 or 4?
- I wonder if the tensor contraction in TRAC leads to higher latency compared with the LoRA adapter. I noticed an increase in training time with larger tensor orders in Table 11.

---

> ### Author Response · Authors · 2025-11-26
> **Response to Reviewer Afac— Part 1/3**
>
> ### R3-Q1. Relation to freezing and sharing approaches (AFLoRA, ShareLoRA)
>
> *(addressing Weakness 1)*
>
> **A1:** We sincerely thank the reviewer for highlighting the connection between TRAC and prior freezing‑based methods (e.g., AFLoRA) and sharing‑based methods (e.g., ShareLoRA). TRAC aims to establish a more fine‑grained and continuously controllable PEFT framework, rather than serving as a simple combination of the two paradigms.
>
> Existing approaches typically control LoRA matrices $A$ and $B$ at the whole‑matrix level. For example, AFLoRA completely freezes matrices $A$ and $B$ during the later training stage, while ShareLoRA enforces layer‑wise sharing across LoRA modules. Both rely on fixed, discrete decisions—either *fully frozen* or *fully shared*—without the ability to smoothly adjust between these extremes, which may cause performance degradation when the shared or frozen scope becomes too large.
>
> To overcome these issues, TRAC applies Tensor‑Train (TT) decomposition, which factorizes each LoRA matrix into several 3‑way tensor cores and enables controlled freezing and sharing at the tensor‑core level:
>
> - **independent cores** retain layer‑specific flexibility;
> - **frozen cores** leverage the inherent redundancy and noise in fine‑tuning;
> - **shared cores** capture cross‑layer commonalities;
> - lightweight vector **controllers** dynamically modulate each layer’s activation over shared cores.
>
> Furthermore, the shape and TT‑ranks of each core can be flexibly adjusted, allowing **continuous control over parameter allocation across modules and layers** compared to the matrix‑level strategies. Empirically, TRAC achieves **lower trainable parameter ratios** than both AFLoRA and ShareLoRA, further advancing the compression frontier of PEFT methods.
>
> We will explicitly include citations to AFLoRA and ShareLoRA in the revised version to better acknowledge their contributions and clarify TRAC’s extension beyond them.
>
> ### R3-Q2. Motivation for pursuing higher parameter efficiency beyond LoRA
>
> *(addressing Weakness 2)*
>
> **A2:** We sincerely thank the reviewer for raising this thoughtful question. We fully understand the concern that standard LoRA already offers a substantial reduction in training cost for *single run*. Our motivation for exploring further compression arises from both **exploratory value** and **practical considerations**, as outlined below.
>
> 1. **Exploratory perspective — understanding fine‑tuning behavior under extreme compression.**
>    In existing LoRA formulations, the lower limit of trainable parameters is constrained by the matrix rank ($r \ge 1$), making it difficult to systematically study how performance scales with parameter reduction. TRAC introduces a Tensor‑Train based PEFT method coupled with cross‑layer sharing, allowing continuous control over tensor shape, rank, and freeze/share ratios. This flexibility provides a new lens for analyzing the trade‑off between *compression and expressivity* in PEFT frameworks.
>
> 2. **Practical perspective — scalability to large‑scale, multi‑task, and personalized deployment.**
>    In real‑world settings, organizations may maintain thousands or millions of downstream adapters for user‑specific or task‑specific fine‑tuning [1]. For example, at the GPT‑3 scale, deploying $10^6$ independent adapters would require hundreds of terabytes of additional storage; **TRAC** could reduce this requirement by roughly 250TB compared to LoRA. Such a gain becomes even more pronounced for larger models. Beyond reducing storage and bandwidth costs, higher compression directly improves the maintainability and update efficiency of massive multi‑version deployments.
>
> In summary, pursuing extreme parameter‑efficient fine‑tuning is valuable both theoretically and practically. We believe TRAC serves as a meaningful step toward this broader exploration.
>
> *[1] Kopiczko, Dawid Jan, Tijmen Blankevoort, and Yuki M. Asano. "VeRA: Vector-based Random Matrix Adaptation." The Twelfth International Conference on Learning Representations.*

---

> ### Author Response · Authors · 2025-11-26
> **Response to Reviewer Afac— Part 2/3**
>
> ### R3-Q3. Intuition behind the symmetry analysis (Appendix A.3)
>
> *(addressing Weakness 4)*
>
> **A3:** We sincerely thank the reviewer for the interest in the symmetry analysis.
>
> In LoRA and its variants, the *symmetry/asymmetry* phenomenon typically refers to the unequal contribution and gradient sensitivity of the two low‑rank matrices $A$ and $B$. Prior works—including LoRA‑FA [2], HydraLoRA [3], ShareLoRA [4], and [5]—have reported similar findings: freezing or sharing the $A$ matrix often leads to only marginal performance change, whereas performing the same on $B$ can cause notable degradation. This suggests that **$B$ plays a more critical role**.
>
> However, standard LoRA imposes a **strictly symmetric low‑rank decomposition**,
>
> $$
> y = W_0x + BAx, \quad \text{where } r_A = r_B = r,
> $$
>
> which constrains both matrices to have identical rank and shape, limiting the ability to capture their intrinsic asymmetry.
>
> **TRAC** breaks this constraint through a Tensor‑Train (TT) decomposition, where the cores corresponding to $A$ and $B$ can be assigned **independent shapes and ranks**. Leveraging the flexibility of TT structures, we can allocate different capacity ratios—for example, $A:B = 2:6$ or $3:5$—instead of a fixed $1:1$ symmetrization. Empirically, we observe that moderately asymmetric configurations yield the best results, while enforcing perfect symmetry can reduce performance.
>
> Intuitively, $A$ can be viewed as an *encoder* that compresses the input $x$ into a low‑dimensional latent representation, and $B$ as a *decoder* that reconstructs this representation back into the output space. The fact that $A$ can be partially shared or frozen supports the hypothesis that more task‑specific adaptation occurs within $B$.
>
> We believe these findings are consistent with observations in previous LoRA‑style works [1-4] and also provide new insight into asymmetric structures in low‑rank parameterization, which may inspire future exploration in this direction.
>
> *[2] Zhang, Longteng, et al. "Lora-fa: Memory-efficient low-rank adaptation for large language models fine-tuning." arXiv preprint arXiv:2308.03303 (2023).*
>
> *[3] Tian, Chunlin, et al. "Hydralora: An asymmetric lora architecture for efficient fine-tuning." Advances in Neural Information Processing Systems 37 (2024): 9565-9584.*
>
> *[4] Song, Yurun, et al. "Sharelora: Parameter efficient and robust large language model fine-tuning via shared low-rank adaptation." arXiv preprint arXiv:2406.10785 (2024).*
>
> *[5] Zhu, Jiacheng, et al. "Asymmetry in Low-Rank Adapters of Foundation Models." International Conference on Machine Learning. PMLR, 2024.*

---

> ### Author Response · Authors · 2025-11-26
> **Response to Reviewer Afac— Part 3/3**
>
> ### R3-Q4. LoRA rank selection and performance fairness
>
> *(addressing Question 1)*
>
> **A4:** We sincerely thank the reviewer for raising this important question regarding the choice of **LoRA rank** and its impact on performance comparison. In our paper, we adopted different LoRA ranks depending on model scale:
>
> - **BERT** (Table 3), **GPT‑2** (Table 5), and **ViT** (Table 6): $r =8$
> - **LLaMA‑2‑7B** (Table 4): $r = 4$.
>
> Our goal was to examine **whether TRAC can maintain—or even exceed—LoRA performance when trained with significantly fewer parameters**. To further evaluate the sensitivity to LoRA rank, we conducted an additional experiment on ViT‑Base (CIFAR‑100) under the same training protocol, varying $r \in \lbrace 1, 2, 4, 8, 16, 32 \rbrace$. The results are summarized below.
>
> ***Table 4. Comparison between LoRA and TRAC under various ranks.***
>
> | **LoRA Rank**              | 1            | 2            | 4            | 8            | 16           | 32           |
> | -------------------------- | ------------ | ------------ | ------------ | ------------ | ------------ | ------------ |
> | **LoRA # Params**          | 36,864       | 73,728       | 147,456      | 294,912      | 589,824      | 1,179,648    |
> | **LoRA Acc. (%)**          | 83.08 (0.38) | 83.31 (0.32) | 83.61 (0.27) | 83.77 (0.26) | 84.01 (0.28) | 84.13 (0.16) |
> | **TRAC # Params**          | 7,488        | 13,632       | 32,832       | 63,552       | 85,824       | 114,240      |
> | **TRAC Acc. (%)**          | 83.17 (0.30) | 83.86 (0.35) | 84.43 (0.25) | 84.42 (0.28) | 84.64 (0.31) | 84.77 (0.25) |
> | **# Params (TRAC / LoRA)** | 20.3 %       | 18.5 %       | 22.3 %       | 21.6 %       | 14.6 %       | 9.7 %        |
> | **Acc. Gap (TRAC – LoRA)** | +0.09        | +0.55        | +0.82        | +0.65        | +0.63        | +0.64        |
>
> **Observations:**
>
> 1. As LoRA rank $r$ increases, both LoRA and TRAC performances improve steadily, showing a consistent scaling trend.
> 2. Across all rank settings, TRAC achieves **+0.09 – +0.82 accuracy gain** while using only **9.7 % – 22.3 % of LoRA’s trainable parameters**.
> 3. Even with just 32,832 trainable parameters (≈ 1/36 of LoRA’s 1.18 M parameters), TRAC outperforms LoRA trained with the highest rank.
>
> Overall, even with higher LoRA ranks, TRAC continues to achieve better accuracy with substantially fewer parameters. Going forward, we plan to extend this analysis to diverse models and datasets for a more comprehensive comparison in future work.
>
> ### R3-Q5. Latency impact of tensor contraction compared to LoRA
>
> *(addressing Question 2)*
>
> **A5:** We sincerely thank the reviewer for the careful observation and insightful question. Indeed, TRAC introduces additional tensor decomposition and contraction operations on top of LoRA, which adds a small amount of computational cost. However, because *matrix multiplications in the backbone network dominate the total FLOPs*, this overhead contributes only **a negligible portion of the overall computation** during both forward and backward passes.
>
> To validate this assessment, we quantitatively compared the total training time of LoRA and TRAC under identical experimental settings. The results are summarized below:
>
> ***Table 5. Training time comparison between LoRA and TRAC (s).***
>
> | Model             | LoRA (s) | TRAC (s) | Relative Overhead |
> | :---------------- | :------: | :------: | :---------------: |
> | **RoBERTa Base**  |  629.2   |  665.3   |      +5.7 %       |
> | **RoBERTa Large** |  1932.3  |  2002.5  |      +3.6 %       |
>
> As shown above, TRAC increases the total training time by only ≈ 3–6%, remaining in the same computational order as LoRA.
>
> Moreover, during inference, TRAC can be **merged into the pretrained weights** just like LoRA, introducing **no additional latency or memory overhead** at deployment.

---

> > ### Comment · Reviewer_Afac · 2025-11-26
> >
> > Thank you for the authors’ response. After careful consideration, I still have concerns about the overall novelty of the work and the extent of improvement compared with LoRA. Therefore, I will maintain my current score.

---

> ### Author Response · Authors · 2025-11-27
> **Response to Reviewer Afac**
>
> We sincerely thank the reviewer for the careful review of our work and for raising concerns about the novelty and the extent of improvement over LoRA.
>
> To further address this point, we have conducted additional experiments comparing our method with LoRA on more recent models and benchmarks.
>
> The new results, summarized below, show that our method consistently achieves better performance while requiring a smaller trainable‑parameter budget.
>
> ### R3-Q6. Additional comparison with LoRA on modern architectures and benchmarks
>
> **A6:** We conducted comparison experiments using settings adopted from LoRA‑GA [6].
>
> - **Models:** LLaMA3‑8B, Qwen3-8B
> - **Commonsense reasoning:** Trained on Commonsense170K and evaluated on eight standard reasoning benchmarks.
> - **Math:** Trained on the MetaMathQA‑100K subset and evaluated on GSM8K.
> - **Code:** Trained on the Code‑Feedback‑100K subset and evaluated on HumanEval.
>
> The additional results are summarized below:
>
> ***Table  6‑A. Commonsense Reasoning Results of LoRA and TRAC (LLaMA3‑8B).***
>
> | Method         | # Param. | BoolQ     | PIQA      | SIQA      | HellaSwag | WinoGrande | ARC-Easy  | ARC-Challenge | OpenBookQA | Avg.      |
> | -------------- | -------- | --------- | --------- | --------- | --------- | ---------- | --------- | ------------- | ---------- | --------- |
> | **LoRA**       | 1.704M   | 69.39     | 85.31     | 76.00     | 90.77     | 77.27      | **89.44** | **76.37**     | **79.00**  | 80.44     |
> | **TRAC Small** | 0.657M   | 70.52     | **85.69** | 75.90     | 91.29     | **81.45**  | 87.84     | 76.28         | 78.40      | 80.92     |
> | **TRAC Large** | 1.001M   | **71.77** | **85.69** | **77.02** | **91.81** | 81.29      | 88.05     | 76.02         | **79.00**  | **81.33** |
>
> ***Table  6‑B. Commonsense Reasoning Results of LoRA and TRAC (Qwen3-8B).***
>
> | Method         | # Param. | BoolQ     | PIQA      | SIQA      | HellaSwag | WinoGrande | ARC-Easy  | ARC-Challenge | OpenBookQA | Avg.      |
> | -------------- | -------- | --------- | --------- | --------- | --------- | ---------- | --------- | ------------- | ---------- | --------- |
> | **LoRA**       | 1.917M   | 67.31     | **88.79** | 79.38     | 90.26     | 76.16      | **96.38** | **90.87**     | **89.40**  | 84.82     |
> | **TRAC Small** | 0.674M   | 66.67     | 87.49     | 79.63     | 92.16     | 80.27      | 96.17     | 90.44         | 88.40      | 85.15     |
> | **TRAC Large** | 1.061M   | **70.95** | 88.74     | **79.99** | **92.37** | **80.98**  | 96.34     | 90.61         | 89.20      | **86.15** |
>
> ***Table  6‑C. Math and Code results of LoRA and TRAC (LLaMA3‑8B).***
>
> | Method         | # Param. | GSM8K     | HumanEval |
> | -------------- | -------- | --------- | --------- |
> | **LoRA**       | 1.704M   | 64.67     | 38.78     |
> | **TRAC Small** | 0.657M   | 63.31     | 38.29     |
> | **TRAC Large** | 1.001M   | **64.90** | **38.90** |
>
> Across all tasks, TRAC achieves **comparable or even superior performance** to LoRA while using only **35.2% – 58.7% of its trainable parameters**.
> In particular, TRAC‑Large improves the **average reasoning accuracy** by **+0.89** on **LLaMA3‑8B** and **+1.33** on **Qwen3‑8B**, while achieving gains of **+0.23** on **GSM8K** and **+0.12** on **HumanEval**.
> These results demonstrate that TRAC maintains parameter efficiency and consistent generalization across modern models and tasks.
>
> We greatly appreciate the reviewer’s helpful and thoughtful comment. We plan to conduct a more systematic and comprehensive evaluation in future work to further validate the generality and applicability of TRAC.
>
> *[6] Wang, Shaowen, Linxi Yu, and Jian Li. "Lora-ga: Low-rank adaptation with gradient approximation." Advances in Neural Information Processing Systems 37 (2024): 54905-54931.*

---

### Official Review · Reviewer_kh71 · 2025-10-31

**Soundness:** 2
**Presentation:** 3
**Contribution:** 1
**Rating:** 0
**Confidence:** 5

**Summary:**

This paper proposes TRAC (Tensor-Train based Across-Layer Compression), a parameter-efficient fine-tuning (PEFT) framework that combines Tensor-Train (TT) decomposition with cross-layer core sharing and freezing. Each LoRA module is represented as a sequence of TT cores (trainable, frozen, shared), and lightweight controllers are added to modulate shared cores per layer. The method is evaluated on BERT, LLaMA2, GPT-2, and ViT models across GLUE, SuperGLUE, E2E, and image classification benchmarks. The authors report comparable or slightly better performance than LoRA and LoRETTA while using 10–20× fewer trainable parameters.

While the method is clearly presented and theoretically consistent, the core idea is NOT novel. Similar hybrid parameterizations have been explored extensively in prior work-most notably COMPACTER (Karimi Mahabadi et al., NeurIPS 2021)—which achieves comparable or better results than full finetuning while training less than 0.05% of parameters, versus This approach substantially higher ratio (12-14%). Moreover, the evaluation is also limited to simple benchmarks, without results on reasoning, math, or code tasks that are now standard for assessing LLM generalization, benchmarks like MMLU, MMLU-pro, humaneval, MBPP, GSM8k, ...

Overall, the contribution is not novel and not more efficient than prior work (compared to Neurips 2021), and the empirical evidence is insufficient to establish this method as a meaningful advance over existing parameter-efficient training methods.

**Strengths:**

The paper is well-written with detailed method descriptions, and consistent notation.
The proposed TRAC framework is technically sound.

The authors conduct comprehensive experiments on multiple architectures (BERT, GPT-2, LLaMA2, ViT) and datasets (GLUE, SuperGLUE, E2E, and image classification), demonstrating that TRAC achieves similar or slightly better performance than LoRA and LoRETTA with fewer trainable parameters. However, these benchmarks are rather outdated and evaluations on more recent benchmarks such as MMLU, MMLU Pro, humaneval, MBPP, GSM8k, ... are needed.

**Weaknesses:**

* Comparison to prior work. The paper fails to acknowledge or compare against COMPACTER (Karimi Mahabadi et al., NeurIPS 2021), which is a foundational and much stronger baseline. COMPACTER achieved similar or better results while fine-tuning only 0.047% of parameters, whereas TRAC trains several orders of magnitude more parameters to reach comparable performance. This omission significantly weakens the contribution and demonstrates limited awareness of prior literature.

* Lack of novelty and missing citations. The paper provides no substantial innovation beyond prior PEFT work. Its main components—Tensor-Train (TT) decomposition and cross-layer parameter sharing-have already been introduced in TT-LoRA (Anjum et al., 2024), LoRETTA (Yang et al., 2024), VeRA (Kopiczko et al., 2024), NoLA (Koohpayegani et al., 2024), and SharedAdapter (Poth et al., 2023). TRAC simply combines these established ideas without offering new theoretical insights or mechanisms.

- Questionable efficiency gains. Although TRAC reduces trainable parameters relative to LoRA, it is unclear whether this translates into any real compute or time savings. The paper does not report FLOPs, memory footprint, or wall-clock improvements.

- Weak empirical results. Gains over LoRA and LoRETTA are small and often within noise. Important baselines such as COMPACTER, and GaLore (https://arxiv.org/pdf/2403.03507) are missing, leaving the evaluation incomplete.

Limited experimental scope. The paper only tests on small or mid-size models (BERT, GPT-2, LLaMA2-7B) and saturated benchmarks (GLUE, SuperGLUE, ImageNet). There is no evaluation on modern reasoning, math, or code benchmarks (e.g., MMLU, GSM8K, MBPP, HumanEval), which are standard in current LLM research.

Overall assessment. TRAC appears to be an incremental reimplementation of prior TT-based and cross-layer methods, presented without proper acknowledgment of earlier work or competitive results. Its contribution is neither novel nor impactful compared to established baselines, particularly COMPACTER (Karimi Mahabadi et al., 2021).

**Questions:**

* Comparison with COMPACTER, and GaLore: how does the method compare in terms of performance and also #trainable parameters compared to existing work of compacter and GaLore? Without these, it is difficult to assess real performance advantages.

* Parameter efficiency vs. compute efficiency: While TRAC reduces trainable parameters, how does it affect FLOPs, memory usage, or wall-clock training time? Can you provide quantitative data on actual resource savings, if any?

* Evaluation scope: Do you plan to evaluate TRAC on modern reasoning, math, and code benchmarks (e.g., MMLU, GSM8K, MBPP, HumanEval)? Such tasks are now standard for assessing LLMs and would strengthen the empirical contribution.

---

> ### Author Response · Authors · 2025-11-26
> **Response to Reviewer kh71 — Part 1/3**
>
> ### R2-Q1. Comparison with  COMPACTER  and novelty clarification
>
> *(addressing Summary, Weaknesses 1/4/6, and Question 1)*
>
> **A1:** We thank the reviewer for emphasizing the importance of **COMPACTER** [1]. We acknowledge its influential role in parameter‑efficient fine‑tuning and will explicitly cite and discuss it in the revision.
>
> **Parameter ratio clarification:**
> The reviewer’s “12–14 %” refers to the ratio *relative to LoRA parameters*, while COMPACTER’s 0.047 % is measured *against the full model size*. Normalized on the same basis, TRAC trains an even smaller fraction—0.021 % for RoBERTa‑Base and 0.0016 % for LLaMA‑2‑7B.
>
> **Methodological distinction:**
> COMPACTER belongs to the **adapter family**, inserting additional nonlinear modules via **Kronecker products**, which modifies the model architecture and requires these modules to be executed during inference. In contrast, TRAC follows the **LoRA line**, applying **tensor‑train (TT) decomposition** in the weight space. The TT modules can be merged into the base weights before inference, avoiding any extra computation or latency. While COMPACTER also explores cross‑layer parameter sharing, TRAC introduces selective sharing with partial freezing and a lightweight vector controller to preserve layer‑specific expressivity within the TT framework.
>
> **Empirical comparison:**
> To further validate these distinctions, we performed a direct comparison between **TRAC** and **COMPACTER** using the *official COMPACTER codebase* under the same experimental setup as [1], with **T5‑Base** evaluated on **GLUE tasks**.
>
> **Table 5. Comparison between COMPACTER and TRAC**
>
> | Method                 |   # Param. |       Ratio |     SST‑2 |      MRPC |      CoLA |      QNLI |       RTE |     STS‑B |      Avg. |
> | ---------------------- | ---------: | ----------: | --------: | --------: | --------: | --------: | --------: | --------: | --------: |
> | COMPACTER [1]          |    104,704 |     0.047 % |     93.81 | **90.69** | **61.27** |     93.08 |     74.82 |     90.46 |     84.02 |
> | COMPACTER (Reproduced) |    104,704 |     0.047 % |     94.04 |     89.71 |     58.56 | **93.30** | **82.01** |     90.33 |     84.66 |
> | **TRAC**               | **34,176** | **0.015 %** | **94.61** | **90.69** |     59.85 |     93.10 | **82.01** | **91.06** | **85.22** |
>
> As shown above, **TRAC** achieves comparable or better performance with only **about 33 % of COMPACTER’s trainable parameters**, indicating a substantially higher degree of parameter efficiency and compression.
>
> Overall, **TRAC** explores a *LoRA‑based* direction within PEFT, extending tensorized modeling along a different technical line from *adapter‑based* methods such as COMPACTER, and further **extends the parameter‑compression frontier** within the PEFT framework.
>
> *[1] Karimi Mahabadi, Rabeeh, James Henderson, and Sebastian Ruder. "Compacter: Efficient low-rank hypercomplex adapter layers." Advances in neural information processing systems 34 (2021): 1022-1035.*
>
> ### R2-Q2. Compute and memory efficiency (FLOPs, wall‑clock time, memory)
>
> *(addressing Weakness 3 and Question 2\)*
>
> **A2:** We sincerely thank the reviewer for raising these valuable questions regarding training time and resource efficiency. We added detailed comparisons against main baselines (LoRA, VeRA, LoRETTA) on both the **RoBERTa** series and **LLaMA‑2‑13B** models. The evaluation covers total training time, peak GPU memory usage during training, and final weight storage size.
>
> ***Table 2‑A. Training time comparison across methods (s).***
>
> | Model         |  LoRA  |  VeRA  | LoRETTA |  TRAC  |
> | :------------ | :----: | :----: | :-----: | :----: |
> | RoBERTa Base  | 629.2  | 814.4  |  697.3  | 665.3  |
> | RoBERTa Large | 1932.3 | 2397.5 | 2121.2  | 2002.5 |
>
> The results indicate that TRAC increases total training time by only ≈5.7 % (Base) and 3.6 % (Large) compared to LoRA, suggesting that **the overhead introduced by tensor decomposition is negligible** in practice.
>
> ***Table 2‑B. Peak GPU memory usage during training (GB).***
>
> | Model         |  LoRA  |  VeRA  | LoRETTA |  TRAC  |
> | :------------ | :----: | :----: | :-----: | :----: |
> | RoBERTa Large | 2.759  | 2.791  |  2.745  | 2.743  |
> | LLaMA‑2‑13B   | 26.571 | 26.731 | 26.649  | 26.605 |
>
> Under identical batch size and optimizer configurations, TRAC and LoRA show nearly identical GPU memory footprints.
>
> ***Table 2-C. Model weight storage size (MB).***
>
> | Model         |  LoRA  | VeRA  | LoRETTA | TRAC  |
> | :------------ | :----: | :---: | :-----: | :---: |
> | RoBERTa Large | 3.014  | 0.389 |  0.658  | 0.197 |
> | LLaMA‑2‑13B   | 12.543 | 1.661 |  2.286  | 0.699 |
>
> The results show that TRAC reduces model storage requirements by approximately **93.5 % (RoBERTa Large)** and **94.4 % (LLaMA‑2‑13B)** compared with LoRA, highlighting its **strong advantage in parameter compression and storage efficiency**.

---

> ### Author Response · Authors · 2025-11-26
> **Response to Reviewer kh71 — Part 2/3**
>
> ### R2-Q3. Comparison with GaLore
>
> *(addressing Weakness 4 and Question 1)*
>
> **A3:** We thank the reviewer for highlighting the relevance of **GaLore** [2]. GaLore is an influential work focusing on training‑time memory efficiency, using low‑rank projections in the *gradient space* to **reduce optimizer‑state storage**. In contrast, TRAC belongs to the LoRA‑family parameter‑efficient tuning methods, operating in the *weight space* to minimize **the number of trainable and storable parameters**.
>
> Because GaLore updates all model weights and does not freeze pretrained parameters, its final model still requires storing the full weights—making its **final storage cost comparable to that of the base model**. TRAC, however, stores only small adapter modules (typically 0.01–0.1 % of model parameters). Thus, the two methods pursue orthogonal efficiency objectives—optimizer‑memory reduction vs. parameter compactness—and are better viewed as complementary rather than directly comparable.
>
> We appreciate this suggestion and will clarify these distinctions and add a discussion of GaLore in the revision.
>
> *[2] Zhao, Jiawei, et al. "GaLore: Memory-Efficient LLM Training by Gradient Low-Rank Projection." International Conference on Machine Learning. PMLR, 2024.*
>
> ### R2-Q4. Novelty and related-work
>
> *(addressing Weakness 4 and Weakness 6)*
>
> **A4:** We sincerely thank the reviewer for raising these valuable comments on the novelty and connections to prior work. While TRAC follows the general philosophy of improving parameter efficiency proposed in earlier studies, its design objectives and implementation differ substantially in several key aspects:
>
> 1. **Tensorized parameterization design under extreme compression.**
>    Methods such as VeRA primarily focus on freezing or sharing matrix‑form adapters, whereas TRAC systematically extends this principle to a Tensor‑Train structure. This extension requires careful choices of tensor order, shape, and decomposition rank—all of which critically affect parameter distribution and representational capacity. TRAC introduces a practical tensorization scheme that enables flexible allocation of ranks and parameters across modules, thereby achieving finer‑grained control over model capacity.
>
> 2. **Cross‑layer sharing with partial freezing.**
>    Previous TT‑based methods typically apply tensor decomposition independently within each layer. In contrast, TRAC develops a unified cross‑layer sharing mechanism that combines shared and partially frozen tensor cores with a lightweight vector controller. The controller modulates layer‑specific variations on top of shared cores, effectively mitigating the degradation observed in naive parameter sharing.
>
> 3. **Asymmetric parameter allocation and scalability.**
>    Leveraging the flexibility of the TT structure, TRAC supports *asymmetric* rank and parameter allocation across different decomposition dimensions, as detailed in Appendix A.3. Under the same parameter budget, assigning a larger TT‑rank to the tensor cores corresponding to the LoRA‑style \(B\) matrix often yields better empirical performance. This mechanism, to our knowledge, has not been systematically explored in prior tensor‑based PEFT studies.
>
> Overall, the innovation of TRAC lies in **advancing the parameter‑efficiency frontier** of PEFT methods by systematically integrating extreme tensorized compression with stable cross‑layer sharing. By coupling shared/frozen tensor cores with controllable vector modulation, TRAC achieves **fewer trainable parameters** than prior methods without sacrificing performance. It retains the **same computational complexity** as LoRA while providing finer‑grained parameter control and improved scalability.
>
> We will further expand the related‑work discussion in the revised version and explicitly clarify TRAC’s contributions in terms of **extreme parameter compression and cross‑layer representation**.

---

> ### Author Response · Authors · 2025-11-27
> **Response to Reviewer kh71 — Part 3/3**
>
> ### R2-Q5. Evaluation on modern reasoning, math, and code benchmarks
>
> *(addressing Summary, Strengths, Weakness 5, and Question 3)*
>
> **A5:** We sincerely thank the reviewer for the insightful and constructive comment regarding the scope of our experiments. We have added experiments on more recent models and tasks, with settings adopted from LoRA‑GA [3].
>
> - **Models:** LLaMA3‑8B, Qwen3-8B
> - **Commonsense reasoning:** Trained on Commonsense170K and evaluated on eight standard reasoning benchmarks.
> - **Math:** Trained on the MetaMathQA‑100K subset and evaluated on GSM8K.
> - **Code:** Trained on the Code‑Feedback‑100K subset and evaluated on HumanEval.
>
> The additional results are summarized below:
>
> ***Table  5‑A. Commonsense Reasoning Results of LoRA and TRAC (LLaMA3‑8B).***
>
> | Method         | # Param. | BoolQ     | PIQA      | SIQA      | HellaSwag | WinoGrande | ARC-Easy  | ARC-Challenge | OpenBookQA | Avg.      |
> | -------------- | -------- | --------- | --------- | --------- | --------- | ---------- | --------- | ------------- | ---------- | --------- |
> | **LoRA**       | 1.704M   | 69.39     | 85.31     | 76.00     | 90.77     | 77.27      | **89.44** | **76.37**     | **79.00**  | 80.44     |
> | **TRAC Small** | 0.657M   | 70.52     | **85.69** | 75.90     | 91.29     | **81.45**  | 87.84     | 76.28         | 78.40      | 80.92     |
> | **TRAC Large** | 1.001M   | **71.77** | **85.69** | **77.02** | **91.81** | 81.29      | 88.05     | 76.02         | **79.00**  | **81.33** |
>
> ***Table  5‑B. Commonsense Reasoning Results of LoRA and TRAC (Qwen3-8B).***
>
> | Method         | # Param. | BoolQ     | PIQA      | SIQA      | HellaSwag | WinoGrande | ARC-Easy  | ARC-Challenge | OpenBookQA | Avg.      |
> | -------------- | -------- | --------- | --------- | --------- | --------- | ---------- | --------- | ------------- | ---------- | --------- |
> | **LoRA**       | 1.917M   | 67.31     | **88.79** | 79.38     | 90.26     | 76.16      | **96.38** | **90.87**     | **89.40**  | 84.82     |
> | **TRAC Small** | 0.674M   | 66.67     | 87.49     | 79.63     | 92.16     | 80.27      | 96.17     | 90.44         | 88.40      | 85.15     |
> | **TRAC Large** | 1.061M   | **70.95** | 88.74     | **79.99** | **92.37** | **80.98**  | 96.34     | 90.61         | 89.20      | **86.15** |
>
> ***Table  5‑C. Math and Code results of LoRA and TRAC (LLaMA3‑8B).***
>
> | Method         | # Param. | GSM8K     | HumanEval |
> | -------------- | -------- | --------- | --------- |
> | **LoRA**       | 1.704M   | 64.67     | 38.78     |
> | **TRAC Small** | 0.657M   | 63.31     | 38.29     |
> | **TRAC Large** | 1.001M   | **64.90** | **38.90** |
>
> Across all tasks, TRAC achieves **comparable or even superior performance** to LoRA while using only **35.2% – 58.7% of its trainable parameters**.
> In particular, TRAC‑Large improves the **average reasoning accuracy** by **+0.89** on **LLaMA3‑8B** and **+1.33** on **Qwen3‑8B**, while achieving gains of **+0.23** on **GSM8K** and **+0.12** on **HumanEval**.
> These results demonstrate that TRAC maintains parameter efficiency and consistent generalization across modern models and tasks.
>
> We greatly appreciate the reviewer’s helpful and thoughtful suggestion. We plan to conduct a more systematic and comprehensive evaluation in future work to further validate the generality and applicability of TRAC.
>
> *[3] Wang, Shaowen, Linxi Yu, and Jian Li. "Lora-ga: Low-rank adaptation with gradient approximation." Advances in Neural Information Processing Systems 37 (2024): 54905-54931.*

---

### Official Review · Reviewer_vXG5 · 2025-11-04

**Soundness:** 2
**Presentation:** 3
**Contribution:** 2
**Rating:** 2
**Confidence:** 4

**Summary:**

This paper proposed a new parameter-efficient adaptation method for transformers. It uses tensor decompositions to target the extreme end of low-paramaters and further reduces the number of necessary parameters by sharing across layers.

**Strengths:**

* easy to understand and simple new PEFT method
* compares well to SOTA methods on several benchmarks while being more parameter-efficient
* releted works section is thorough and gives good overview

**Weaknesses:**

* the computational complexity analysis is a bit unclear. I assume this is for a) for training time and b) for the number of adapted parameters (this should be made more clear). But then VeRA's complexity seems wrong, as there's no n^2 complexity in its decompositions. How is this arrived at and what even is "computational order"?
* The importance of the theoretical analysis is unclear. The fact that more parameters -> more expressivity is trivial and the fact that sharing across layers can be used too. Does this yield any insights into how to use the adapter or how it should be designed? Otherwise this seems like an appendix without a use.
* the PEFT method builds on existing works such as LoReTTA and several works that rely on frozen random matrices
* old models. By now, llama-2 and GPT2 benchmarks have become irrelevant and newer models should be used with correspondingly adapted benchmarks. examples include qwen, llama-3 and or vision-language models.
* No mention of training times
* minor: use citations with \citep when referring to a paper

**Questions:**

* given a thorough hyperparameter evaluation protocol, can the authors provide results on more recent architectures such as llama-3?
* provide clarity on the use of the theoretical results and why they are included
* Provide clarity on "computational order"
* Provide insights on training time & memory required for training experiments.

---

> ### Author Response · Authors · 2025-11-26
> **Response to Reviewer vXG5 — Part 1/3**
>
> ### R1-Q1. Clarification on computational complexity (“computational order”)
>
> *(addressing Weakness 1 and Question 3)*
>
> **A1:** We thank the reviewer for the insightful comments. In our work, *computational order* refers to the **dominant order** of computational cost in the **forward pass of the PEFT module**. For instance, in LoRA the model output can be written as  $y = W_0x + \alpha B A x $, and we focus on the computational cost of the second term $BAx$. Below, we provide the complete derivation and explanation.
>
> Assuming the model has $L$ layers, each parameter matrix $W \in \mathbb{R}^{n \times n}$, and denoting the low-rank dimension (or tensor TT rank) as $r$, the forward complexity for different methods is as follows:
>
> 1. **LoRA:**
>    Sequentially computing $Ax$ and $B(Ax)$ yields about $O(4nr)$ per layer, giving an overall complexity of $O(Lnr)$.
>
> 2. **VeRA:**
>    In practice, the selected rank $r_{\text{VeRA}}$ is typically large (e.g., $r_{\text{VeRA}} = 1024$ in both BERT and GPT-2). In this case, it can be approximated that $r_{\text{VeRA}} \approx n$, leading to a per-layer cost of $O(n r_{\text{VeRA}}) \approx O(n^2)$. This approximation is consistent with the empirical trend observed in our experiments (see Table in A4), where VeRA training requires longer time than LoRA, LoRETTA, and TRAC.
>
> 3. **Tensor Train methods (LoRETTA / TRAC):**
>    Based on the fast tensor multiplication algorithm described in Appendix B.4, the per-layer complexity is approximately $O(nr + n^{2/3}r^2 + n^{1/3}r^2) \approx O(nr)$, resulting in a total complexity of $O(Lnr)$.
>
> We will clarify the exact definition of *computational order* and include the complete theoretical derivation in the revised version to improve clarity and precision.
>
> ### R1-Q2. Use and value of the theoretical analysis
>
> *(addressing Weakness 2 and Question 2)*
>
> **A2:** We appreciate the reviewer’s valuable comments. We acknowledge that the theoretical results are not meant to provide new discoveries, but rather to offer an **intuitive, design‑motivated explanation** for TRAC.
>
> The analysis connects tensor rank with expressivity and shows that sharing tensor cores can retain approximation ability under certain assumptions—providing insight into why TRAC’s shared structure is effective within a small fixed parameter budget.
>
> Overall, the theory serves a heuristic and interpretive role rather than a formal guarantee. We will clarify this motivation and also plan to develop a more systematic theoretical framework in future work to further understand tensor methods in PEFT.
>
> ### R1-Q3. Relation to prior PEFT methods (LoRETTA and frozen‑matrix approaches)
>
> *(addressing Weakness 3)*
>
> **A3:** We sincerely thank the reviewer for pointing out the connections between TRAC and prior PEFT approaches. While TRAC follows the general philosophy of parameter efficiency established in earlier works, our method introduces **distinct structural and algorithmic advances** in its representation form, sharing strategy, and implementation design.
>
> 1. **Tensorized parameterization design under extreme compression.**
>    Extending a matrix‑based method like LoRA to higher‑order tensor representations requires careful specification of tensor order, shape, and decomposition rank—all of which critically influence parameter distribution and model expressivity. TRAC proposes a practical tensorization scheme that enables flexible allocation of ranks and parameters across modules, achieving finer‑grained control over model capacity.
>
> 2. **Cross‑layer sharing framework with partial freezing.**
>    Existing TT‑based methods (e.g., LoRETTA) typically apply tensor decomposition independently within each layer. In contrast, TRAC develops a unified framework for cross‑layer core sharing and partial freezing, combined with a lightweight vector controller that preserves layer‑specific diversity atop shared tensor cores. This effectively prevents the degradation commonly observed in naive parameter sharing.
>
> 3. **Asymmetric parameter allocation and scalability.**
>    Leveraging the flexibility of the TT structure, TRAC supports asymmetric rank and parameter allocation across different decomposition dimensions, as detailed in Appendix A.3. Given the same total parameter budget, assigning a larger TT‑rank to the tensor cores corresponding to the LoRA‑style $B$ matrix often yields better empirical performance.
>
> Overall, TRAC **advances the parameter‑efficiency frontier** of PEFT methods, achieving **fewer trainable parameters** than prior approaches without sacrificing performance. It also retains the **same $O(nr)$ computational complexity** as LoRA while enabling finer‑grained parameter control and effective cross‑layer sharing.

---

> ### Author Response · Authors · 2025-11-26
> **Response to Reviewer vXG5 — Part 2/3**
>
> ### R1-Q4. Training time and memory efficiency evaluation
>
> *(addressing Weakness 5 and Question 4)*
>
> **A4:** We sincerely thank the reviewer for the attention to training time and resource efficiency. We include detailed comparisons with main baselines (LoRA, VeRA, LoRETTA) on both the **RoBERTa** series and **LLaMA‑2‑13B** models, covering training time, peak GPU memory usage during training, and final model storage size.
>
> ***Table  4‑A. Training time comparison across methods (s).***
>
> | Model         |  LoRA  |  VeRA  | LoRETTA |  TRAC  |
> | :------------ | :----: | :----: | :-----: | :----: |
> | RoBERTa Base  | 629.2  | 814.4  |  697.3  | 665.3  |
> | RoBERTa Large | 1932.3 | 2397.5 | 2121.2  | 2002.5 |
>
> The results indicate that TRAC increases total training time by only ≈ 5.7% (Base) and 3.6% (Large) compared to LoRA, suggesting that **the overhead introduced by tensor decomposition is negligible**.
>
> ***Table  4‑B. Peak GPU memory usage during training (GB).***
>
> | Model         |  LoRA  |  VeRA  | LoRETTA |  TRAC  |
> | :------------ | :----: | :----: | :-----: | :----: |
> | RoBERTa Large | 2.759  | 2.791  |  2.745  | 2.743  |
> | LLaMA‑2‑13B   | 26.571 | 26.731 | 26.649  | 26.605 |
>
> Under identical batch size and optimizer configurations, TRAC and LoRA exhibit nearly identical GPU memory footprints.
>
> ***Table  4‑C. Model weight storage size (MB).***
>
> | Model         |  LoRA  | VeRA  | LoRETTA | TRAC  |
> | :------------ | :----: | :---: | :-----: | :---: |
> | RoBERTa Large | 3.014  | 0.389 |  0.658  | 0.197 |
> | LLaMA‑2‑13B   | 12.543 | 1.661 |  2.286  | 0.699 |
>
> The results demonstrate that TRAC reduces model storage costs by approximately **93.5%** (on RoBERTa Large) and **94.4%** (on LLaMA‑2‑13B) compared with LoRA, reinforcing its **strong advantage in parameter compression**.
>
> ### R1-Q5. Citation formatting (use of `\citep`)
>
> *(addressing Weakness 6)*
>
> **A5:** We thank the reviewer for carefully pointing out this formatting issue. We will thoroughly check all references in the revised version and ensure that every paper citation consistently uses the `\citep{}` format.

---

> ### Author Response · Authors · 2025-11-27
> **Response to Reviewer vXG5 — Part 3/3**
>
> ### R1-Q6. Updating experiments to modern models and benchmarks
>
> *(addressing Weakness 4 and Question 1)*
>
> **A6:** We sincerely thank the reviewer for the insightful and constructive comment regarding the scope of our experiments. We have added experiments on more recent models and tasks, with settings adopted from LoRA‑GA [1].
>
> - **Models:** LLaMA3‑8B, Qwen3-8B
> - **Commonsense reasoning:** Trained on Commonsense170K and evaluated on eight standard reasoning benchmarks.
> - **Math:** Trained on the MetaMathQA‑100K subset and evaluated on GSM8K.
> - **Code:** Trained on the Code‑Feedback‑100K subset and evaluated on HumanEval.
>
> The additional results are summarized below:
>
> ***Table  6‑A. Commonsense Reasoning Results of LoRA and TRAC (LLaMA3‑8B).***
>
> | Method         | # Param. | BoolQ     | PIQA      | SIQA      | HellaSwag | WinoGrande | ARC-Easy  | ARC-Challenge | OpenBookQA | Avg.      |
> | -------------- | -------- | --------- | --------- | --------- | --------- | ---------- | --------- | ------------- | ---------- | --------- |
> | **LoRA**       | 1.704M   | 69.39     | 85.31     | 76.00     | 90.77     | 77.27      | **89.44** | **76.37**     | **79.00**  | 80.44     |
> | **TRAC Small** | 0.657M   | 70.52     | **85.69** | 75.90     | 91.29     | **81.45**  | 87.84     | 76.28         | 78.40      | 80.92     |
> | **TRAC Large** | 1.001M   | **71.77** | **85.69** | **77.02** | **91.81** | 81.29      | 88.05     | 76.02         | **79.00**  | **81.33** |
>
> ***Table  6‑B. Commonsense Reasoning Results of LoRA and TRAC (Qwen3-8B).***
>
> | Method         | # Param. | BoolQ     | PIQA      | SIQA      | HellaSwag | WinoGrande | ARC-Easy  | ARC-Challenge | OpenBookQA | Avg.      |
> | -------------- | -------- | --------- | --------- | --------- | --------- | ---------- | --------- | ------------- | ---------- | --------- |
> | **LoRA**       | 1.917M   | 67.31     | **88.79** | 79.38     | 90.26     | 76.16      | **96.38** | **90.87**     | **89.40**  | 84.82     |
> | **TRAC Small** | 0.674M   | 66.67     | 87.49     | 79.63     | 92.16     | 80.27      | 96.17     | 90.44         | 88.40      | 85.15     |
> | **TRAC Large** | 1.061M   | **70.95** | 88.74     | **79.99** | **92.37** | **80.98**  | 96.34     | 90.61         | 89.20      | **86.15** |
>
> ***Table  6‑C. Math and Code results of LoRA and TRAC (LLaMA3‑8B).***
>
> | Method         | # Param. | GSM8K     | HumanEval |
> | -------------- | -------- | --------- | --------- |
> | **LoRA**       | 1.704M   | 64.67     | 38.78     |
> | **TRAC Small** | 0.657M   | 63.31     | 38.29     |
> | **TRAC Large** | 1.001M   | **64.90** | **38.90** |
>
> Across all tasks, TRAC achieves **comparable or even superior performance** to LoRA while using only **35.2% – 58.7% of its trainable parameters**.
> In particular, TRAC‑Large improves the **average reasoning accuracy** by **+0.89** on **LLaMA3‑8B** and **+1.33** on **Qwen3‑8B**, while achieving gains of **+0.23** on **GSM8K** and **+0.12** on **HumanEval**.
> These results demonstrate that TRAC maintains parameter efficiency and consistent generalization across modern models and tasks.
>
> We greatly appreciate the reviewer’s helpful and thoughtful suggestion. We plan to conduct a more systematic and comprehensive evaluation in future work to further validate the generality and applicability of TRAC, and to extend it to broader model families.
>
> *[1] Wang, Shaowen, Linxi Yu, and Jian Li. "Lora-ga: Low-rank adaptation with gradient approximation." Advances in Neural Information Processing Systems 37 (2024): 54905-54931.*

---

### Author Response · Authors · 2025-12-03
**Rebuttal Summary — Part 2/2**

### 5. Clarifications on Reviewer‑Specific Points

- **R1‑Q1 — Clarification on computational complexity:**
  We have provided a precise definition and complete derivation of the forward‑pass complexity for TRAC and the baselines. The analysis confirms that TRAC’s computational cost is on par with LoRA, consistent with our experimental observations.

- **R1‑Q2 — Role of the theoretical analysis:**
  The theoretical analysis is intended to provide an intuitive, design‑driven interpretation of TRAC’s effectiveness rather than to introduce a new formal theory.

- **R3‑Q3 — On asymmetry:**
  The two low‑rank matrices in LoRA play unequal roles, with $B$ contributing more to adaptation. TRAC exploits this asymmetry through fine‑grained, non‑uniform rank allocation, achieving more effective parameter usage and stronger efficiency under the same budget.

- **R3‑Q4 — Effect of increasing LoRA rank:**
  Experiments varying the LoRA rank show that both methods scale smoothly with larger parameter budgets, while TRAC consistently delivers higher accuracy using only about 20% of LoRA’s parameters, confirming the robustness of its efficiency advantage.

- **R4‑Q1 — Vector controller and activation function:**
  The lightweight vector controller enhances the cross‑layer adaptability of shared cores with negligible parameter overhead. Motivated by MoE‑style gating, we adopt a softmax activation, which ablation studies confirm as the most effective choice.

- **R4‑Q3 — Choice of tensor order, shape, and rank:**
  We analyzed how tensor configurations affect performance under a fixed parameter budget:
  - **Order:** Third‑order tensors achieve a strong trade‑off between simplicity and accuracy; higher orders show no severe degradation.
  - **Shape:** Keeping dimensions balanced ($n_1 \approx n_2 \approx n_3$) yields stable results.
  - **Rank:** Start from an equal‑rank initialization and adjust as needed to allocate capacity.

### Overall Summary

During the rebuttal stage, we systematically addressed the **reviewers’ main concerns** through targeted analyses and additional experiments on recent models and benchmarks.
The new results consistently support **TRAC’s design**—integrating tensor‑based freezing and sharing with lightweight control—and demonstrate **stable efficiency and performance** across models and tasks.
Taken together, these clarifications and findings provide a clear and balanced view of the method’s rationale, implementation, and empirical behavior within the PEFT framework.

We hope these responses have effectively clarified the raised questions and helped convey the **technical soundness and contributions** of our work.
We sincerely thank the reviewers and the area chair for their time and constructive feedback.

---

### Author Response · Authors · 2025-12-03
**Rebuttal Summary — Part 1/2**

We sincerely thank the reviewers and the area chair for their careful reading and insightful feedback.

Their thoughtful comments have greatly helped us refine the presentation, deepen our analysis, and conduct new experiments that further clarify the contributions of **TRAC**.
During the rebuttal stage, we carefully and substantially addressed the main concerns and performed additional studies to directly respond to the specific questions raised.

We summarize below the key responses and new supporting evidence.

### 1. Novelty and Comparison with Prior Work

*(Details in R1‑Q3, R2‑Q4, R3‑Q1)*

While earlier PEFT methods have explored either *parameter sharing* or *parameter freezing*, they primarily focused on one strategy at a time, whereas **TRAC jointly leverages both** within a **unified tensor‑train decomposition framework**. Building on this unified design, we further develop the following key technical innovations:

1. A **practical and flexible framework** for configuring tensor representations, enabling fine‑grained and efficient integration of parameter freezing and sharing.
2. A **layer‑wise lightweight vector controller** that flexibly modulates shared cores, improving adaptability with minimal overhead.
3. An **asymmetric allocation scheme** leveraging the *different functional roles* of LoRA’s $A$ and $B$ matrices to enhance efficiency under a fixed‑budget setting.

Across diverse models and datasets, **TRAC** consistently matches or surpasses existing PEFT baselines—including LoRA and other sharing‑, freezing‑, and tensor‑based methods—while requiring **substantially fewer trainable parameters**.

### 2. Experiments on Modern Models and Datasets

*(Details in R1‑Q6, R2‑Q5, R3‑Q6)*

To strengthen the empirical evidence and address reviewers’ concerns, we extended the evaluation to the **Llama‑3‑8B** and **Qwen3-8B** models and several modern benchmarks, including:

- **Commonsense reasoning:** Trained on Commonsense170K and evaluated on eight standard reasoning benchmarks.
- **Math:** Trained on MetaMathQA and evaluated on GSM8K.
- **Code:** Trained on Code‑Feedback and evaluated on HumanEval.

These new experiments show that TRAC achieves **comparable or superior performance** than LoRA while using only **35.2 % – 58.7 %** of its trainable parameters.
Notably, TRAC‑Large improves the **average reasoning accuracy** by **+0.89** on **LLaMA3‑8B** and **+1.33** on **Qwen3‑8B**, with consistent gains also observed on **GSM8K** and **HumanEval**, demonstrating its strong parameter efficiency and generalization on modern models.

### 3. Computational Cost, GPU Memory, and Storage

*(Details in R1‑Q4, R2‑Q2, R3‑Q5)*

We conducted additional quantitative comparisons on training time, GPU memory usage, and storage cost against the main baselines. The results indicate that:

- The **training‑time overhead** of TRAC is *below 6%* relative to LoRA, confirming that the cost introduced by tensor decomposition is negligible.
- **Peak GPU memory usage** during training remains nearly identical.
- The **model weight storage** is reduced by *over 93%*.

Overall, these results demonstrate that TRAC delivers a **substantial advantage in parameter compression** with **minimal computational overhead**.

### 4. Clarification and Comparison with COMPACTER

*(Details in R2‑Q1, R4‑Q6)*

The comparison raised by Reviewer R2‑kh71 primarily concerns **adapter‑based** methods such as COMPACTER, whereas **TRAC** is developed within the **LoRA‑based** paradigm—two *parallel technical lines* in PEFT research.

TRAC employs **Tensor‑Train decomposition** with **selective freezing, cross‑layer core sharing,** and **lightweight controllers** that can be **merged into the pretrained weights** during inference. This design preserves the original model architecture and introduces no inference‑time overhead.

In contrast, **COMPACTER** relies on partially shared **Kronecker‑based adapter modules** that remain active during inference, resulting in additional computational and memory overhead at deployment.

Furthermore, we conducted direct experiments comparing **TRAC** and **COMPACTER** under identical setups. The results show that **TRAC achieves comparable or better performance while using only about 33% of COMPACTER’s trainable parameters**, demonstrating its higher degree of parameter efficiency and compression.

These methodological and empirical distinctions establish **TRAC** as an **independent LoRA‑line approach** rather than a variant of existing adapter methods such as COMPACTER, and further **extend the parameter‑compression frontier** within the PEFT framework.

---

### Meta-Review · Area_Chair_xcBv · 2026-01-02

**Summary:**

Reviewer vXG5 gave a reject (2) score, citing the lack of new models for experiments, lack of mention of training time, and asked for clarifications regarding complexity. The authors responded sufficiently.

Reviewer kh71 gave a particularly low score (0), and mentioned that there were missing baselines, lack of report of efficiency metrics, and no experimentation on reasoning/math/code benchmarks. The authors responded comprehensively.

Reviewer Afac responded to keep their score (6) , while sk7c (8) was well addressed.

Personally, I would predict the score to be higher if a "normal" rebuttal happened. If the reviewers were responsive, I would think the score could be at least 4268 (i.e., average 5). If the reviewers were responsible, the average could be higher than 5. As such, I recommend an accept. However, if the SAC thinks otherwise, I wouldn't mind if the decision is bumped down.

**Reviewer Concerns:**

I believe vXG5 and kh71's concerns were well addressed.

**Reviewer Scores:**

I believe vXG5 and kh71's concerns were well addressed. Hence, if they are "humane" or responsive enough, they would have raised the score to at least 4 and 2 respectively, or more. I personally think the rebuttal was decent, and would have raised the score more if I were them, but we all know  from experience of the trend with reviewers being unwilling to raise scores two notches above their initial scores. Hence, I will be conservative and predict that the final score could look like 4286, i.e., average of 5.

---

### Decision · Program_Chairs · 2026-01-26

Accept (Poster)